# Cell membrane glycan contents are biochemical factors that constitute a kinetic barrier to viral particle uptake in a protein-nonspecific manner

**Yoshihisa Kaizuka\*, Rika Machida**

National Institute for Materials Science, Tsukuba, Japan

## eLife Assessment

This **fundamental** work substantially advances our understanding of how the glycocalyx of cells provide a non-specific barrier for the interaction of viruses with cell-surface receptors. Using both in vitro experiments and in vivo manipulations they provide **compelling** evidence for the properties of the glycocalyx to serve as an energy barrier as a main attribute of its mode of action. The work will be of broad interest to virologists and the cell biology community that studies host-pathogen interactions.

**\*For correspondence:**
KAIZUKA.Yoshihisa@nims.go.jp

**Competing interest:** The authors declare that no competing interests exist.

**Abstract** Various types of glycoproteins have been suggested to inhibit viral infection of cells via steric repulsion. However, it is difficult to evaluate such physical actions genetically, simply because they are nonspecific and can be caused by any molecule. Therefore, we investigated a method to compare this nonspecific action among cells with diverse membrane protein profiles. We found that a wide range of glycoproteins individually had a strong inhibitory effect on infection, while on the other hand, the total amount of glycans was negatively correlated with the infection level in each cell. Thus, the infection-inhibitory effect of glycoproteins was molecularly nonspecific but was additively enhanced according to the amount of glycans on the cell surface. In this correlation, glycans function as a fundamental factor. Further investigating the mechanism by which glycans function as a factor in infection control, we conclude that the repulsion between proteins created by branched glycans forms a kinetic energy barrier against packing the virus into the region of protein interstitial space. As a result, the formation of the adhesive interface between the virus and the cell membrane, which is necessary for infection, is inhibited. This study attempted to link the cell's nonspecific physical properties with intracellular biochemicals. A similar approach may be applied to quantify other nonspecific biological phenomena.

## Introduction

Numerous types of physical and mechanical factors of cells, both intrinsic mechanical properties such as stiffness and external mechanical stimuli such as osmotic pressure, have been reported as regulators of important biological events at the cellular level (*Delgado and Cabernard, 2020*; *Du et al., 2023*; *Enyedi et al., 2016*; *Jalihal et al., 2020*; *Saraswathibhatla et al., 2023*). Some of these events are regulated by a few important genes, while others are controlled by multiple factors and are considered gene-nonspecific events. As one such physical event, we are studying steric inhibition of viral infection by bulky membrane proteins. Studies in the Covid-19 pandemic have highlighted the fact that, as has long been recognized in other infectious diseases, even infection

by the same viral strain can have highly variable consequences for host disease progression. To analyze such diverse host-side responses in Covid-19, intensive high-throughput studies have been conducted, accumulating vast amounts of data at the cellular, organ, or individual patient level (*Ahern et al., 2022*; *Biering et al., 2022*; *Delorey et al., 2021*; *Ellinghaus et al., 2020*; *Grant et al., 2021*; *Hadjadj et al., 2020*; *Meckiff et al., 2020*; *Ravindra et al., 2021*; *Yoshida et al., 2022*; *Ziegler et al., 2021*). As a result, various host-side factors have been proposed. Particularly, innate immune molecules such as various cytokines have received significant attention, but several others have also been reported. Mucin-related glycoproteins are among the latter group, as suggested by cellular CRISPR screens as well as by genome-wide association studies (*Biering et al., 2022*). There have also been many studies at the molecular, cellular, or animal level on the function of these highly glycosylated proteins in viral infection, including not only mucin but also other proteins including PSGL1 and CD43 (*Chatterjee et al., 2023*; *Delaveris et al., 2020*; *Linden et al., 2008*; *McAuley et al., 2017*; *Murakami et al., 2020*; *Wardzala et al., 2022*). While there are known examples of glycans that function as viral receptors (*Thompson et al., 2019*), these results demonstrate that a variety of glycoproteins negatively regulate viral infection in a wide range of systems. These glycoprotein groups have no common amino acid sequences or domains. The glycans modified by these proteins include both N-glycans conjugated with asparagine and O-glycans conjugated with serine and threonine. N- and O-glycans are each composed of many types of monosaccharides and form complex Glycome as a whole (*Dworkin et al., 2022*; *Huang et al., 2021*; *Moremen et al., 2012*; *Rademacher et al., 1988*). The association of specific glycan structures with various diseases has been studied, but there have been no reports of such specificity with regard to the inhibition of viral infection. All of these results suggest that bulky membrane glycoproteins nonspecifically inhibit viral infection.

If the protein size is comparable to the size of the virus and the protein density is high, it is conceivable that the bulky glycoproteins could act as a physical barrier to viral uptake on the cell membrane. However, it is not easy to estimate and compare the steric inhibitory effects of these membrane proteins on different cells with very diverse expression profiles. If glycoproteins affect viral infection in a nonspecific manner, and if this is due to steric inhibition as suggested (*Delaveris et al., 2020*; *Murakami et al., 2020*) then polymers without glycans could have the same function. Indeed, synthetic polymers that bind to cell membranes have been shown to modulate viral infection in a manner dependent on molecular size and molecular density (*Kaizuka and Machida, 2023*; *Pyrć et al., 2021*). Inferring from this fact, it is possible that viral infection in cells is regulated by the expression profile and density of all membrane proteins.

Here, we attempt to biochemically describe this nonspecific steric effect on the cell surface. For this purpose, we combine biochemical dissection for individual molecules with physical analyses. With physical modeling alone, the phenomenon might be described by a small number of parameters, but these parameters cannot be easily associated with genes. On the other hand, it is not straightforward to estimate the steric effects of an entire cell by measuring and summing all the effects of individual genes. To combine these two approaches, we attempted to find biochemical factors that relate steric effects across the cell surface to information of biochemical components in the cell. Such factors could link the physical quantity of steric effects with genomic or transcriptomic data of the cell. Previous studies showing steric effects of glycoproteins suggested that glycans may be a fundamental biochemical factor regulating such an overall phenomenon.

To confirm that glycans are a general chemical factor of steric repulsion, an extensive list of glycoproteins on the cell membrane surface would be useful. The wider the range of proteins to be measured, the better. Therefore, we collect information on glycoproteins on the genome and compile them into a list that is easy to use for various purposes. Then, by analyzing sample molecules selected from this list, it may be possible to infer the effect of the entire glycoprotein population on the steric inhibition of virus infection, despite the complexity and diversity of the Glycome (*Dworkin et al., 2022*; *Huang et al., 2021*; *Moremen et al., 2012*; *Rademacher et al., 1988*). Elucidation of the mechanism of how glycans regulate steric repulsion will also be useful to quantitatively discuss the relationship between steric repulsion and intracellular molecular composition. For this purpose, we apply the theories of polymer physics and interface chemistry.

## Results

### List of membrane glycoproteins in human genome and their inhibitory effect on virus infection

To test the hypothesis that glycans contribute to steric repulsion at the cell surface, we first generate a list of glycoproteins in the human genome and then measure the glycan content and inhibitory effect on viral infection of test proteins selected from the list (*Figure 1A*). To compile the list of glycoproteins, we first selected all human membrane proteins in the database (UniProt) based on the presence of a transmembrane domain or membrane anchor domain and found that 81.5% (2515 molecules) of these membrane proteins had previously been reported or predicted to be glycosylated (*Bateman et al., 2023*). We then used two machine learning-based software packages (NetNGlyc/NetOGlyc and GlycoEP) to determine whether amino acid residues in the ectodomain of all 2515 molecules are glycosylated (*Figure 1B*, *Supplementary file 1* and *Supplementary file 2*; *Chauhan et al., 2013*; *Gupta and Brunak, 2002*; *Steentoft et al., 2013*). Both software packages predicted one or more O- or N-glycosylation in 2441 and 2290 proteins, respectively, of which 2226 molecules (88.5% of the total) were predicted by both software packages (*Figure 1B, C*, *Figure 1—figure supplement 1A*). The number of highly glycosylated molecules for which more than 15% of the amino acids in the extracellular domain were predicted to be glycosylated was 67 and 75, respectively, of which 45 were common to both software packages (*Figure 1B–C*). The Pearson correlation coefficient between the predicted glycosylation rates of all molecules calculated by each software was 0.656 (*Figure 1—figure supplement 1B*). Thus, the predictions of the two software programs were in relatively good agreement, with relatively few molecules predicted to be highly glycosylated throughout the human genome. Both software programs also detected three times as many sequences predicted to undergo O-type glycosylation as those predicted to undergo N-type glycosylation (3.08 and 3.00 times more, respectively), and this trend was particularly pronounced for highly glycosylated molecules. Overall, many of the listed highly glycosylated molecules are consistent with molecules already known to be glycosylated, suggesting the consistency of our prediction strategy. It is also likely that some of the protein sequences analyzed in this study may be identical to the sequences used in machine learning during software development.

To evaluate the predicted results, the glycosylation of 18 sample molecules selected from the list was measured. These sample molecules include MUC1 and its truncation mutants, other types of highly glycosylated molecules, and receptor and ligand molecules involved in cell-cell interactions. Using HEK293T cell lines exogenously expressing genes of these proteins tagged with fluorescent markers, their glycosylation was measured by binding of a lectin from Arachis hypogaea (PNA), and the number of these proteins in the cells was measured simultaneously. PNA was used for the measurement because it has a wider dynamic range than other lectins (*Figure 1—figure supplement 1C*). This suggests that GalNAc recognized by PNA is predominantly present on glycans of HEK293T cells, especially on the termini of glycans that are amenable to lectin binding, compared to other saccharides. Flow cytometry measurements showed a nearly linear relationship between the signals of bound lectins and membrane protein (*Figure 1D*, *Figure 1—figure supplement 1D*). This relationship can be explained by a first-order reaction model of lectin binding to populations of cells with diverse protein expression levels (see Methods). The model also predicted that, when the slope PNA/mol obtained from this linear regression is compared among proteins, it should have a linear relationship with the level of glycosylation of each protein. We assumed that the total glycosylation level per molecule should reflect the number of glycosylation sites that were predicted by the software. Then, we evaluated the correlation between the experimentally determined PNA/mol and the predicted number of glycosylation sites for each protein and found that the two were indeed in a linear relationship (*Figure 1E*). Therefore, we can conclude that within these sample molecules, the number of potential glycosylation sites predicted by the software correlates well with the actual amount of glycans for each molecule.

Although our experiments were performed only in a specific cell type, the measured amount of glycans per molecule correlated well with the number of predicted amino acid sites to be glycosylated. Especially for highly glycosylated proteins, the agreement was good. On the other hand, the measured values for low glycosylation molecules were more strongly influenced by the background glycans of endogenous proteins. This robustness in prediction may result from the sequence diversity of glycosylation sites. When the total amount of glycosylation in each protein is compared as in our

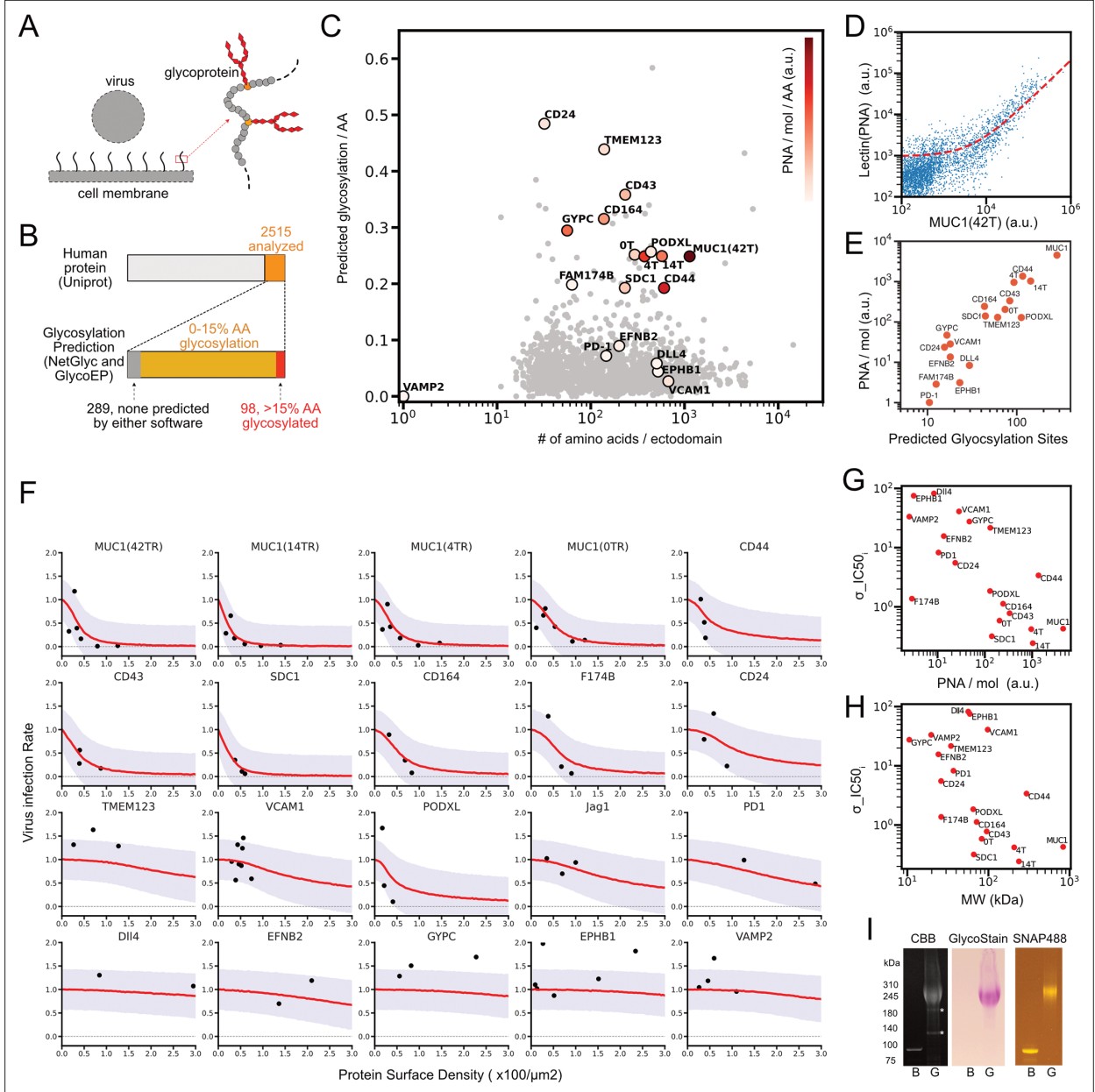

**Figure 1.** List of glycoproteins in the human genome: predictions and measurement of glycosylation, and virus infection inhibition assays using sample proteins from the list. (**A**) Schematic of inhibition of virus infection by membrane glycoproteins. (**B**) The number of membrane proteins and predicted glycosylated proteins in human genome from UniProt. (**C**) The number of predicted glycosylation sites per the number of amino acid sequence of ectodomain for 2515 membrane associated proteins, plotted along with the number of ectodomain amino acid sequence. Color indicates the measured rate for glycosylation per molecule (PNA/mol) per amino acid. 0T, 4T, and 14T indicate truncation mutants of MUC1 that contain 0, 4, and 14 tandem repeat sequences, respectively. (**D**) Flow cytogram for the binding of Alexa Fluor 647 labeled PNA to HEK293T cells expressing MUC1(42 tandem repeats) tagged with SNAP surface 488 and the linear regression of the data to the reaction model (red dashed line, see the method section for details). (**E**) Relations of the measured PNA/mol and the number of predicted glycosylation sites for the indicated molecules. (**F**) SARS-CoV2-PP infection assay in HEK293T cells expressing ACE2, TMPRSS2, and each of designated membrane protein. Dots were measured values of the integral of GFP expressions from infected viruses in those samples adjusted by the total ACE2 expressions at the time of infection and were plotted along with the mean density of membrane protein at the time of infection. Red lines indicate learned predicted infection rates mean from Bayesian hierarchical inference based on sigmoidal function, and purple area represents one sigma below and above the red lines. (**G**) Relations between the measured rate for glycosylation per molecule (PNA/mol) and molecular specific IC50 density in sigmoidal inhibitory function inferred from Bayesian hierarchical modeling in F (σ_IC50). (**H**) Relations between σ_IC50 and estimated molecular weight including glycans in the experimental system. (**I**) Purification and analysis of recombinant proteins, non-glycosylated (bacterial or B) and glycosylated (**G**) MUC1 (14TR) tagged with SNAP surface 488. Coomassie Brilliant Blue stained (left), glycan stained (middle), and fluorescent (right) for proteins in SDS-PAGE.

*Figure 1 continued on next page*

*Figure 1 continued*

The online version of this article includes the following figure supplement(s) for figure 1:

**Figure supplement 1.** List of glycoproteins in the human genome: technical details for predictions and measurement of glycosylation, and virus infection inhibition assays using sample proteins from the list.

case, molecules with a wider spectrum of predicted amino acid sequences should have a greater probability of being glycosylated, even if there is a bias in sequences to be glycosylated in each cellular system. Therefore, we propose that our software-based method generated a list of glycoproteins useful for estimating the total glycan content of each protein in various cell types.

We then examined whether these membrane glycoproteins can inhibit viral infection, and whether their inhibitory effects depend on the degree of protein glycosylation. We measured the infection of SARS-CoV2 pseudoparticle virus based on Lentivirus (SARS-CoV2-PP) consisting of the Spike molecule of SARS-CoV2 (Wuhan-Hu-1; *Figure 1F*, *Figure 1—figure supplement 1E*; *Chan et al., 2020*; *Ke et al., 2020*). Cell surface densities of the viral receptor ACE2 and the membrane glycoprotein to be studied were measured prior to viral infection, and the infection was measured after 48 hr (*Figure 1—figure supplement 1E, G*). The results showed that all highly glycosylated proteins, with the exception of GYPC, which has the shortest ectodomain, had an inhibitory effect on infection. These proteins include those previously reported (MUC1, CD43) as well as those not yet reported (CD44, SDC1, CD164, F174B, CD24, PODXL; *Delaveris et al., 2020*; *Murakami et al., 2020*). In contrast, other molecules (VCAM-1, EPHB1, TMEM123, etc.) showed little inhibitory effect on infection within the density range we used. Infection of other types of viruses was similarly inhibited by glycoproteins (*Figure 1—figure supplement 1F*). Previous results showed that membrane-bound synthetic polyethyleneglycols (PEGs) inhibit SARS-CoV-2-PP infection at surface density above 2000 $\mu m^{-2}$ (*Kaizuka and Machida, 2023*). In contrast, the range of mean densities at which highly glycosylated molecules could inhibit infection was an order of magnitude lower, ~100 $\mu m^{-2}$. This result suggests that glycosylation of membrane proteins has a significant impact on the inhibition of virus infection.

To quantitatively compare the infection inhibitory effects of all glycoproteins in a unified manner, a hierarchical Bayesian model was constructed and data from all molecules were regressed on this unified model at once (*Figure 1F*, *Figure 1—figure supplement 1I*). This approach was based on the assumption that each glycoprotein inhibits viral infection by a common mechanism, but the degree of inhibition depends on the protein-specific parameter. As a model for regression, a standard sigmoidal inhibition function was used with the density of glycoproteins as a variable. The parameter IC50 (surface density to achieve 50% inhibition of infection) of the membrane proteins was estimated from the posterior predictive distribution obtained by Markov chain Monte Carlo sampling for this Bayesian model (*Figure 1—figure supplement 1J*). We found that IC50 was inversely correlated with the amount of glycosylation of each molecule (Pearson correlation coefficient, –0.29; *Figure 1G*); IC50 was similarly correlated with the molecular weight including glycans, but was not significantly correlated with amino acid length (correlation coefficient of –0.39 and 0.11, respectively; *Figure 1H*, *Figure 1—figure supplement 1K*). These results indicate that the total amount of glycans has a significant effect on the inhibition of viral infection.

One advantage of using such a hierarchical model is that one can easily obtain the overall trend without making many measurements on all molecules as in ordinary statistics. Thus, it is an effective approach to reduce batch effects in experiments such as ours where many different items need to be compared. Note that due to the nature of Bayesian statistics, the results may differ slightly depending on the prior distribution, and that while the amount of data we had is sufficient for hierarchical inference for all molecules, there may not be enough data for each molecule in analyzing individually in the manner of ordinary statistics.

The molecular weights of the glycans in those proteins were estimated from the data of glycan mass in purified MUC1 ectodomain (*Figure 1I*). Molecular weight for other proteins was estimated by applying the data for lectin binding assays (*Figure 1D–E*). From these data, we can estimate that a typical glycoprotein in our system is glycosylated at ~50% of the predicted sites by assuming the average size of O-glycans as decasaccharides with a molecular weight of ~2000. These results indicate that in our experimental system, glycosylation of proteins increases their molecular weight by only a few folds at most. However, even glycoproteins shorter than 200 amino acids had a significant inhibitory effect on virus infection, while some of longer but low-glycosylated proteins didn't.

## Cellular glyco-populations and infection control

Since a wide range of glycoproteins inhibit viral infection, it is possible that all types of glycoproteins have an additive effect for this function. To test this hypothesis, we infected a monolayer of epithelial cells endogenously expressing highly heterogeneous populations of glycoproteins with SARS-CoV-2-PP and measured viral infection from cell to cell visually by microscope imaging. We found that highly infected cells were spatially exclusive with highly glycan-expressing cells, indicating that there was a clear inverse correlation between the amount of virus infection and the amount of surface-bound lectins in the cells (*Figure 2AB*). In this cell line, this inverse correlation was most pronounced when quantifying N-acetylneuraminic acid (Neu5AC, recognized by lectins SSA and MAL) compared to the various types of glycans, while some other glycans also showed weak correlations (*Figure 2—figure supplement 1C*). These results showed that the amount of virus infection in cell anti-correlated with the amount of total glycans on the cell surface. As the amount of glycans is determined by the total population of glycocalyx, the infection inhibitory effect can be additive by glycoprotein populations, as we hypothesized.

If the inhibitory effect is nonspecific and additive, the contribution of each protein is likely to be less significant. To confirm this, we also measured the correlation between the density of each glycoprotein and viral infection. CD44, which was shown to account for about 10% of all cell surface glycans (*Wyler et al., 2021*), showed a weak inverse correlation with viral infection; even such a weak correlation was not observed with other proteins, including ERBB2, which is approximately four-fold more highly expressed than CD44 (*Figure 2C-D*, *Figure 2—figure supplement 1A-C*). Furthermore, the amount of glycans on the cells did not correlate with the expression of CD44 alone or any other specific protein, suggesting that all the different glycoproteins contributed to the total glycan content in an additive manner (*Figure 2E-F*, *Figure 2—figure supplement 1D*). Intracellular glycosylation levels are further regulated by complicated pathways including various types of glycosyltransferases, which are likely confounding factors in the causal relationship between glycoprotein expression and infection inhibition. Proteins that showed very minor correlations might also be present somewhere in the network of confounding factors. Our results demonstrate that total glycan content is a superior indicator than individual glycoprotein expression for assessing infection inhibition effect generated by cell membrane glycocalyx. These results are consistent with our hypothesis regarding the additive nature of the nonspecific inhibitory effects of each glycoprotein.

Pearson correlation is effective for comparing samples with varying scales of data because it normalizes the data values by the mean and variance. However, as observed in our experiments, this may not be the case when the distribution of data within a sample varies between samples. In addition, as has already been reported, the distribution of infected cells often deviates significantly from the normal distribution of data that is the premise of Pearson correlation (*Russell et al., 2018*; *Figure 2B*). To further analyze data in such nonlinear situations, we applied the threshold overlap score (TOS) analysis (*Figure 2G-H*, *Figure 2—figure supplement 1E*). This is one statistical method for analyzing nonlinear correlations and is specialized for colocalization analysis in dual-color images (*Sheng et al., 2016*). TOS analysis involves segmentation of the data based on signal intensity, as in other nonlinear statistics (*Reshef et al., 2011*). The computed TOS matrix indicates whether the number of objects classified in each region is higher or lower than expected for uniformly distributed data, which reflects co-localization or anti-localization in dual-color imaging data. For example, calculated TOS matrices show strong anti-localization for infection and glycosylation when both signals are high (*Figure 2G–H*). This confirms that high infection is very unlikely to occur in cells that express high levels of glycans. The TOS analysis also yielded better anti-localization scores for some of the individual membrane proteins, especially those that are heterogeneously distributed across cells (*Figure 2H*). This suggests that TOS analysis can highlight the inhibitory function of molecules that are sparsely expressed among cells, reaffirming that high expression of a single type of glycoprotein can create an infection-protective surface in a single cell and that such infection inhibition is not protein-specific. In contrast, for more uniformly distributed proteins such as the viral receptor ACE2, TOS analysis and Pearson correlation showed similar trends, although the two are mathematically different (*Figure 2D and H*). Because glycoprotein expression levels and virus-derived GFP levels were treated symmetrically in these statistical calculations, the same logic can be applied when considering the heterogeneity of infection levels among cells. Therefore, it is expected that TOS analysis can reasonably compare samples with different virus infection level distributions by focusing on cells with high infection levels in all samples.

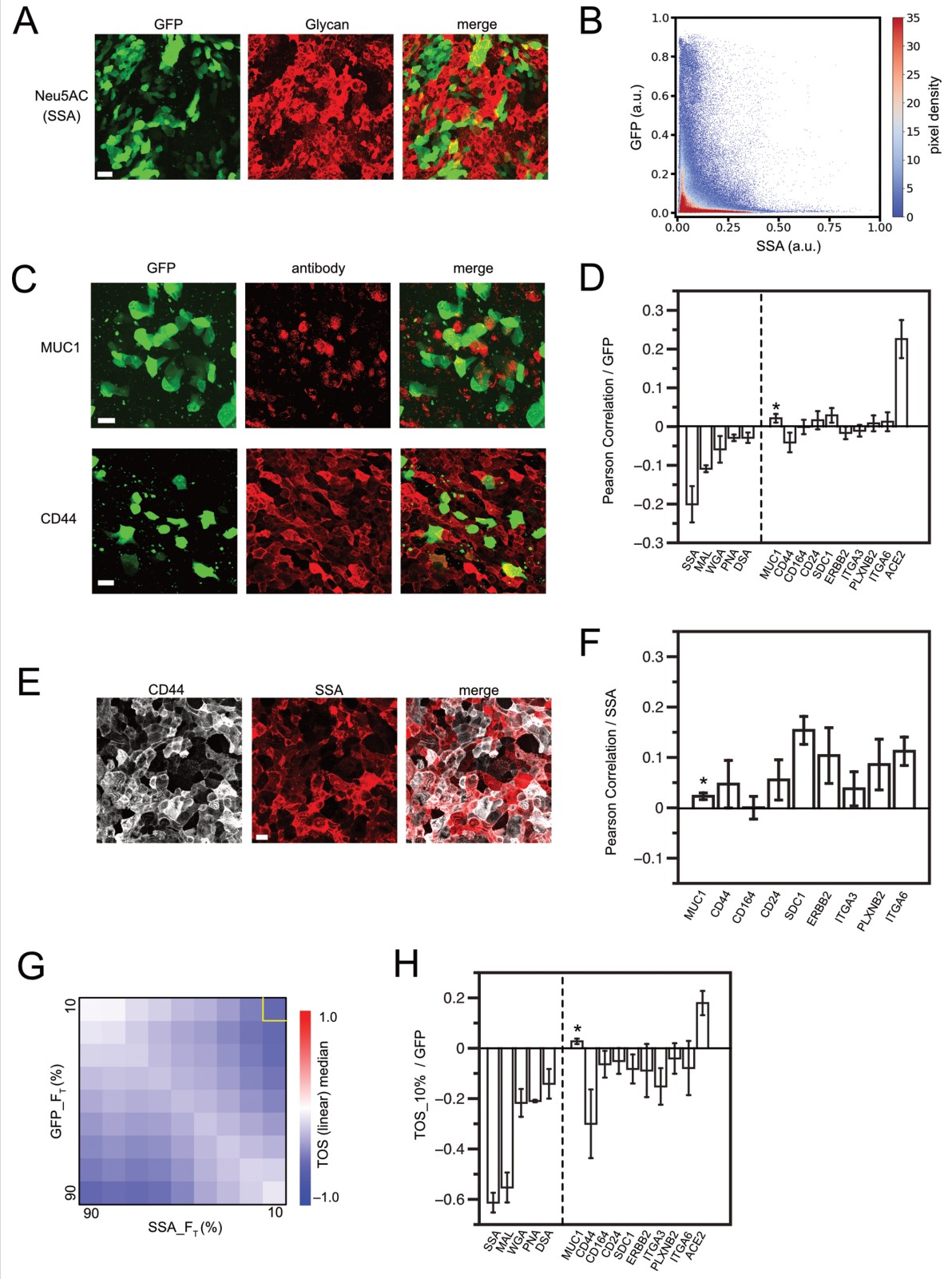

**Figure 2.** Virus infection in epithelium regulated by glycan contents in each cell. (**A**) Maximum projection of Z-stack images at 1 μm intervals taken with a confocal microscope. SARS-CoV2-pp-infected, air-liquid interface (ALI)-cultured Calu-3 cell monolayers were chemically fixed and imaged by binding of Alexa Fluor 647-labeled Neu5AC-specific lectin from Sambucus sieboldiana (SSA) and GFP expression from the infecting virus. (**B**) Density scatter plot of normalized fluorescence intensities in all pixels in (**A**) in both GFP and SSA channels. Color indicates the pixel density. (**C**) SARS-CoV2-

*Figure 2 continued on next page*

*Figure 2 continued*

pp infected Calu-3 ALI monolayer, imaged by immunofluorescence for each membrane protein and GFP. Maximal projection of z-stack of images is shown. (**D**) Pearson correlation for fluorescence intensities of lectin/antibody and GFPs in image pixels in maximal projection of z-stack images. Error bars are standard error of mean from images from three or more different samples. Lectins used were SSA, MAL (from *Maackia amurensis*, Neu5AC specific), WGA (from *Triticum vulgaris*, GlcNAc specific), PNA, and DSA (from *Datura stramonium*, GlcNAc specific). (**E**) SARS-CoV2-pp infected Calu-3 ALI monolayer, imaged by immunofluorescence for CD44 and SSA lectin binding. Maximal projection of z-stack images is shown. (**F**) Pearson correlation of lectin/antibody signal and SSA signal in image pixels in maximal projection of z-stack images. Error bars are standard error of mean from images from three or more different samples. (**G**) TOS analysis for GFP and SSA signal pixels in image of (**A**). (**H**) Correlation in top 10% in both axes in the TOS analysis. Error bars are standard error of mean from images from three or more different samples. *MUC1 has a small mean expression level and variance and is more affected by measurement noise than other molecules when calculating the Pearson correlation function (**C** and **F**). In addition, the number of cells in which expression can be detected is small, so no significant correlation was detected by TOS analysis (**H**). Scale bars = 40 μm (**A**) and 20 μm (**C, E**).

The online version of this article includes the following figure supplement(s) for figure 2:

**Figure supplement 1.** Virus infection in epithelium regulated by glycan contents in each cell.

Our findings suggest that membrane glycoproteins nonspecifically inhibit viral infection, and we hypothesize that their inhibitory function is also nonspecific depending on the type of glycan. Our hypothesis is consistent with the observations in the TOS analysis. Although minor saccharide species in the system (such as GlcNAc and GalNAc recognized by DSA, WGA, or PNA) showed anti-colocalization with infection, their scores were much lower than those of major saccharide species. This suggests that all major and minor saccharide species have an infection inhibitory effect, but cells enriched with minor type glycans are only partially present in the system, and the contribution of these cells to virus inhibition is also partial. It is also consistent with the observation that the amount of GalNAc recognized by PNA determines the virus infection inhibition in HEK 293T cells (*Figure 1*). Therefore, we believe that our assay using a single type of predominantly expressed lectin is still useful for estimating the total glycan content. Nevertheless, the virus infection rate may show a better correlation with a more accurately estimated total glycan in each cell. For example, the use of multiple lectins with appropriate calibration to integrate multiple signals to simultaneously detect a wider range of saccharide species would allow for more accurate estimation. It should be noted that the amount of bound lectin does not necessarily measure the overall glycan composition but likely reflects the sugar population at the free end of the glycan chain to which the lectin binds most.

## Influence of membrane glycoproteins on viral binding or uptake

Glycans could be one of the biochemical substances that link the intracellular molecular composition and macroscopic steric forces at the cell surface. To clarify this connection, we further investigated the mechanism by which membrane glycoproteins inhibit viral infection. First, we measured viral binding to cells to determine which step of infection is inhibited. We found that a large number of SARS-CoV2-PP can still bind to cells even when cells expressed sufficient amounts of the glycoprotein (mean density ~50 μm$^{-2}$) that could account for the majority of glycans within these cells and inhibit viral infection (*Figure 3A*). Similarly, on the two-dimensional culture surface of Calu-3 cells, no negative correlation was observed between the number of viruses bound and the total amount of glycans on the cell surface (*Figure 2—figure supplement 1F–G*). The slight positive correlation between bound virus and glycans may be due to higher expression levels of viral receptors in glycan-rich cells. These results indicate that glycoproteins do not inhibit virus binding to cells, but rather inhibit the steps required for subsequent virus internalization. When cells form the interface with various types of viruses, including SARS-CoV-2, plasma membranes are invaginated for preparing uptake (*Ogando et al., 2020*). At these interfaces, apposed membranes of viruses and cells are tightly adhered to proceed to fusion. Thus, we hypothesize that membrane glycoproteins do not inhibit the initial binding of viruses to cells via point-to-point molecular binding of receptors and ligands, but rather inhibit the formation of stable interface that is required for membrane fusion and viral uptake.

Both of these two inhibition phenomena are mediated by the steric effect of glycoproteins, but the mechanisms are different. To inhibit all cell-virus receptor-ligand binding, a high density of glycoproteins larger than the size of the receptor-ligand complex is required (*Figure 3B*). In contrast, much smaller amounts of shorter proteins would be sufficient to inhibit the formation of the cell-virus interface. In our measurements, a protein with an extracellular domain of ~200 amino acids (e.g. CD164

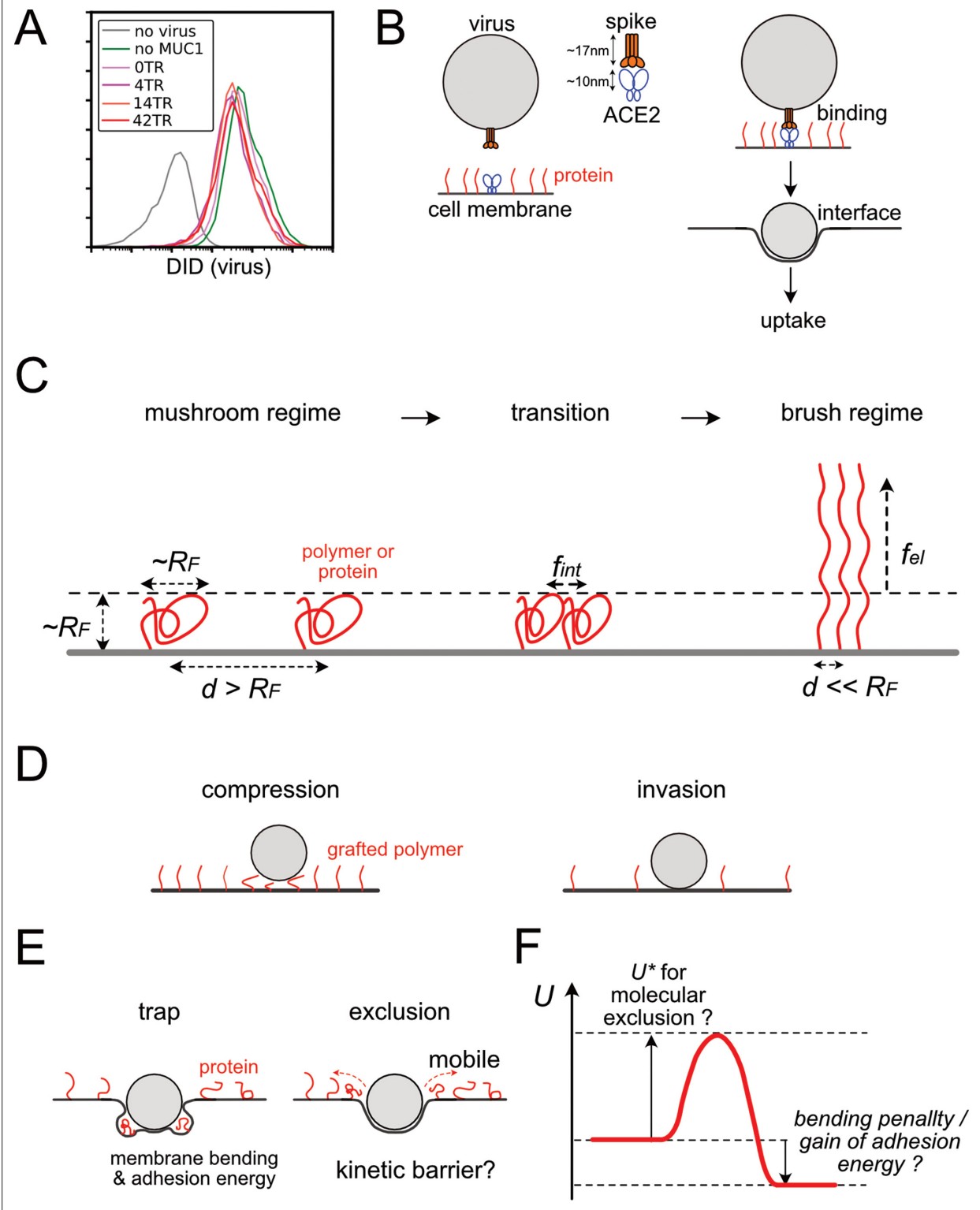

**Figure 3.** Structure of interface between viruses and cell membranes and polymer brush theory. (**A**) Flow cytogram of DID labeled SARS-CoV2-PP binding to HEK293T cells expressing ACE2, TMPRSS2, and/or various truncation mutants of MUC1. (**B**) Schematic diagram of the interface structure between the virus and the cell membrane during the process from cell-virus binding to virus uptake via stable interface formation. (**C**) Basics of polymer brush theory and free energy of the polymer in the course of mushroom to brush transitions. $d$, spacing distance between polymers, $R_F$, Flory radius of polymers, $f_{int}$, and $f_{el}$, intermolecular and elastic free energies per a single polymer. (**D**) Two types of interfacial structures of particle binding to polymer

*Figure 3 continued on next page*

Figure 3 continued

graft surfaces in the conventional polymer brush model. (**E**) Two additional structure types of particle-surface interface, specialized cases for virus-cell interface. (**F**) A chart for free energy of the system *U* during the process of virus-cell interface formation.

The online version of this article includes the following figure supplement(s) for figure 3:

**Figure supplement 1.** Virus binding to cells and fluid phase endocytosis were not significantly affected by surface glycoproteins.

[138aa]) at a density of ~100 $\mu m^{-2}$ showed significant inhibition in viral infection. This molecule is shorter than the receptor ACE2 (722 aa), and its density corresponds to only one protein per 100 nm square area, while the diameter of the virus is also ~100 nm. And even under these conditions, binding of the virus to the cell was not inhibited (***Figure 3A***). Therefore, we investigated how such inhibition of interface formation occurs.

We also investigated the effect of membrane glycoproteins on membrane trafficking, another process involved in viral infection. Expression of MUC1 with higher number of tandem repeats reduced the dextran transport in the fluid phase, while expression of multiple membrane glycoproteins that have infection inhibitory effects, including truncated MUC1 molecules, showed no effect on fluid phase endocytosis, indicating a molecular weight-dependent effect (***Figure 3—figure supplement 1B***). The molecular weight-dependent inhibition of endocytosis may be due to factors such as steric inhibition of the approach of dextran molecules or a reduction in the transportable volume within the endosome. In any case, it is clear that many low molecular weight glycoproteins inhibit infection by disturbing processes other than endocytosis. Based on the above, we focus on the effect of glycoproteins on the formation of the interface between the virus and the cell membrane.

We introduced the framework of conventional polymer brush theory to study the structure of virus-cell interfaces containing proteins. This theory provides a general foundation for considering the behavior of polymers at interfaces (***Alexander, 1977***; ***de Gennes, 1980***; ***Milner, 1991***). In this theory, the polymers on the surface are considered to be in either a 'mushroom' or 'brush' regime, depending on their density (***Figure 3C***). In the mushroom regime, the size of the polymer is estimated to be the mean of the radius of gyration in equilibrium, which is defined as Flory radius ($R_F$). When the surface density becomes very high, the polymer forms a brush, which distorts the polymer and reduces its projected size on the plane. Numerous experimental measurements of the formation of polymer brushes have also been reported (***Overney et al., 1996***; ***Wu et al., 2002***; ***Zhao and Brittain, 2000***). In these measurements, the transition to a brush typically occurs at a density higher than that required to pack a surface with hemispherical polymers of diameter $R_F$. This is the point at which the energy loss due to repulsive forces between adjacent molecules ($f_{int}$) exceeds the energy required to stretch the polymer perpendicularly into a brush ($f_{el}$). Polymer brush theory also applies to cell membrane proteins that diffuse in lipid membranes (***Evans et al., 1996***; ***Hiergeist and Lipowsky, 1996***; ***Shurer et al., 2019***). In particular, since glycoproteins have a bottlebrush structure with branched sugar chains on amino acid chains, our system consists of a brush of bottle brushes (***Van den Steen et al., 1998***).

In the framework of polymer brush, cases of approaching particles to the surface containing graft polymers have been studied. These cases can be classified as either 'invasion' or 'compression' depending on the density of the graft polymer (***Figure 3D***; ***Halperin, 1999***). The case of the virus-cell interface is similar to this previous case but is distinctive because the protein is mobile and the membrane is flexible. Therefore, we further assume that there are two cases: one is 'trap', which involves molecular trapping and deformation of the cell membrane, and the other is 'exclusion', in which molecular exclusion results in the formation of an adhesive interface. (***Figure 3E***). Invasion without involving any molecular exclusion occurs only when the molecular density is very dilute. Molecular exclusion is enabled by molecular fluidity of the cell membrane, and such exclusion has been observed on a macroscopic scale at the cell-cell interface in the immune system or in the apoptotic blebs (***Le et al., 2024***; ***Sibener et al., 2018***; ***Varma et al., 2006***).

Using this framework, we consider the free energy of the virus-cell system. Trapping of membrane proteins at the interface is accompanied by a bending energy penalty or loss of adhesion energy (***Figure 3E***). These should depend on the size and number of trapped proteins. The free energy of the polymers can also be considered; in the mean-field approximation based on Flory polymer theory, the free energy of brush polymers can be decomposed into two main components: the interaction energy between polymers $f_{int}$ and the elastic energy $f_{el}$ of individual polymers. (***Figure 3C***; ***Alexander, 1977***; ***de Gennes, 1980***; ***Halperin, 1999***; ***Li et al., 2022***; ***Milner, 1991***). If the formation

of the cell-virus interface involves macroscopic motion of particles, such as molecular exclusion, the process may experience an energy barrier $U^*$ involved in such transition (*Figure 3F*; *Halperin, 1999*). These arguments about free energy involve several assumptions. For example, it was assumed that when these proteins are trapped at the cell-virus interface, the protein volume and Flory radius are conserved. This is probably because lipid bilayers are flexible and their bending takes less energy than protein compression. This should be supported by a series of previous observations on the structure of proteins at the membrane-membrane interface (*Carbone et al., 2017*; *Paszek et al., 2014*; *Wong and Groves, 2002*). Besides, the processes of virus binding to membrane and membrane invagination were not explicitly included in this discussion because they are common to different types of interfacial structures.

## Effect of glycosylation in protein size and conformation

Protein size can be a key parameter in steric interaction. To measure the size of proteins on the plasma membrane in situ, fluorescence resonance energy transfer (FRET) between donor fluorescent tags at the ectodomain ends of proteins and acceptor fluorescent molecules incorporated into the plasma membrane was measured in living cells using a fluorescence lifetime imaging microscope (FLIM). (*Figure 4A*). In this setting, FLIM-FRET measured time average of end-to-end distance of proteins in solution in equilibrium, which is equivalent to the ensemble average of the radii of gyration of the ectodomain in solution, although one end of these proteins was attached to plasma membrane. Since the size of grafted polymers on solid surfaces measured by FRET was reported to be close to the Flory radius ($R_F$) of these polymers (*Ma et al., 2014*), we considered that the protein size measured by FLIM-FRET in our system also corresponds to $R_F$ of each protein.

We found that the FLIM-FRET data were well explained by the FRET model with a planar membrane geometry in which acceptors are distributed on the lipid membrane in two dimensions (*Figure 4B*; *Gibson and Loew, 1979*; *Wong and Groves, 2002*). We then extended this model to a hierarchical Bayesian model with acceptor density as a variable and applied it to FRET data for all molecules that were measured in the same cell type and with the same fluorescent dye pair. By regressing all measured data on the hierarchical model at once, we obtained $R_F$ for each molecule from the estimated posterior predictive distribution (*Figure 4—figure supplement 1A–B*). The longest estimated $R_F$ that could be resolved was 7.5 nm. Molecules with barely detectable FRET changes (e.g. MUC1 [14TR, 42TR], CD44) were expected to be larger than this maximum measurable distance and were not included in the analysis. Importantly, $R_F$ of some glycoproteins (e.g. CD43) was found to be smaller than the receptor-ligand binding length of ACE2 and Spike (~27 nm), yet they inhibited viral infection. Thus, this result supports the rationale for our interface hypothesis (*Figure 3B*).

The measured $R_F$ was compared to the IC50 of infection inhibition (*Figure 4C*). The two parameters showed a good correlation for many molecules, but there was a group of molecules that deviated from this correlation. Outliers showing exceptionally weak inhibition (e.g. VCAM1, EPHB1) were found to be poorly glycosylated. This suggests that even proteins with similar equilibrium size in solution may have different inhibitory effects on infection depending on the amount of glycosylation. In the case of glycoproteins, there is a complex confounding relationship between these parameters because the number of amino acids to be glycosylated increases as the length of amino acids increases. However, it is suggested that glycosylation affects certain molecular properties other than average molecular size to produce the infection inhibitory effect.

Before examining the direct role of glycans in suppressing infection, the effects of glycans on molecular properties were examined from various perspectives. We found that $R_F$ was strongly correlated with the molecular weight including glycans, but weakly correlated with the number of amino acids (correlation coefficient of 0.87 and 0.58, respectively; *Figure 4D–E*). This observation is consistent with previous studies showing that Flory radii of branched polymers are determined by the total molecular weight (*Hermans et al., 1952*; *Kreussling and Ullman, 1954*). This suggests that $R_F$ reflects the total chain length of amino acids and glycans, for which the total molecular weight is a better measure. It also suggests that for the same amino acid chain length, a protein with more glycans would have a longer $R_F$. This hypothesis can be tested by estimating the scaling exponent $v$ in the Flory model for $R_F \sim N^v$. The chain length $N$ is the number of amino acids in this case. We estimated $v$ separately for highly glycosylated molecules (MUC1, CD43, CD164, CD24) and for low glycosylated molecules. The resulting $v$ for the two groups are 0.144 and 0.108, respectively. This shift

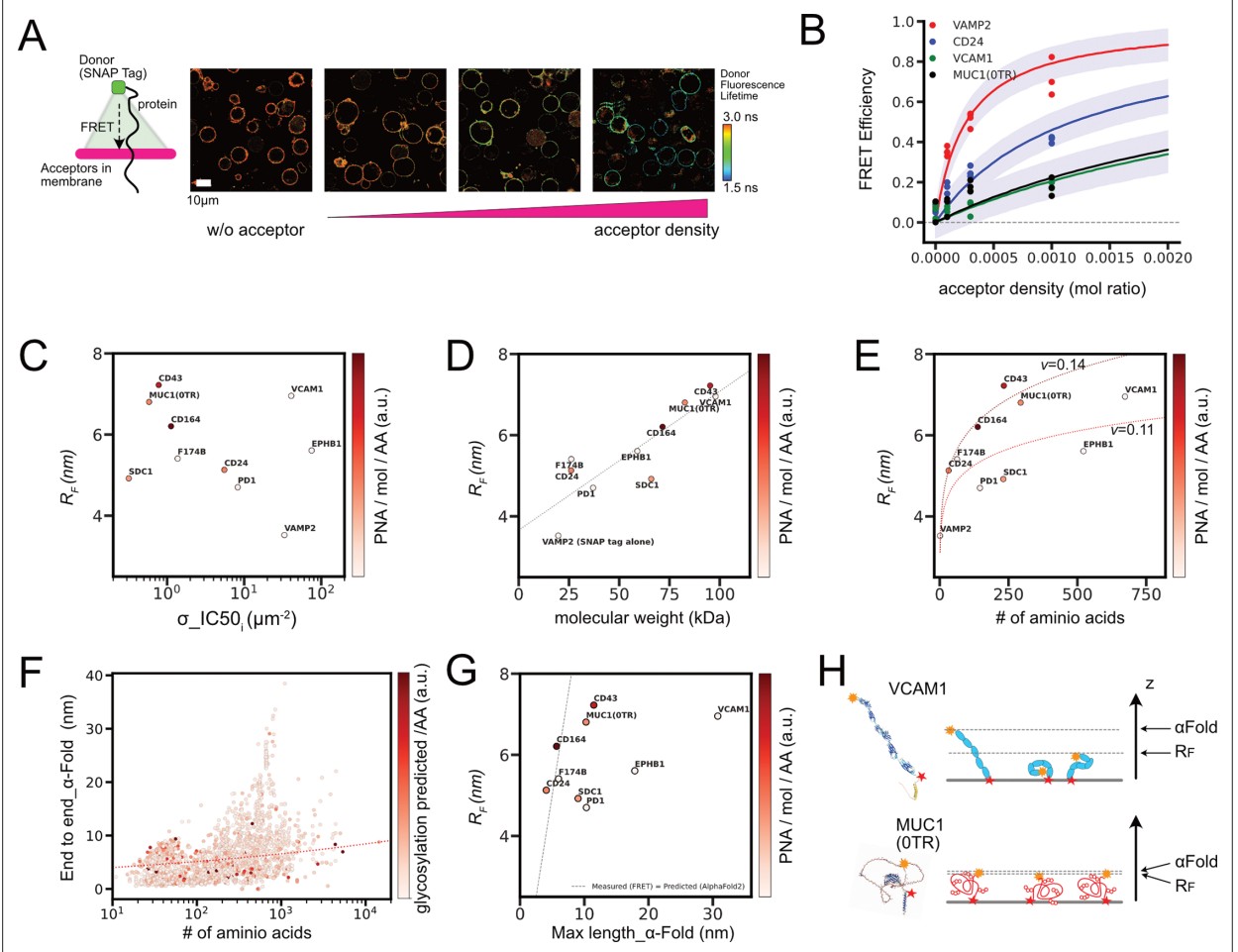

**Figure 4.** In situ FLIM–FRET measurements for protein sizes and conformational predictions. (**A**) Example of FLIM images for cells expressing SNAP– Surface 488 conjugated CD24 proteins and being incorporated with PlasMem Bright Red dyes in different surface densities. Schematic drawing (left) depicts a geometry of the FRET from a single donor dye conjugated to SNAP tag at the end of protein ectodomain to populations of acceptor dyes incorporated in plasma membrane. (**B**) FRET efficiency estimated from FLIM imaging for cells expressing each protein at different mean acceptor densities. Lines are predicted mean FRET efficiencies from Bayesian hierarchical inference (see Method details), and purple area represents one sigma below and above the lines. (**C**) Relations between inferred Flory radius from FLIM – FRET analyses and inferred $\sigma_{IC50}$ from infection inhibition assays. Measured glycosylation rate PNA/mol/AA was depicted in color. (**D**) Relations between inferred Flory radius from FLIM – FRET analyses and estimated molecular weight including glycans. Dot line is the result of linear regression. (**E**) Relations between inferred Flory radius from FLIM – FRET analyses and amino acid length. Dot lines are the fit to the Flory model for $R_F \sim N^\nu$. (**F**) Relations between the distance between coordinates of two amino acids at both ends of ectodomain in Alpha Fold2 predicted conformations and number of amino acids for all 2515 proteins in the list. The number of predicted glycosylation sites per amino acid was depicted in color. The dot line indicates the Flory model $R_F \sim N^\nu$, where $\nu$ =0.14. (**G**) Relations between the distance between coordinates of two amino acids at both ends of ectodomain in Alpha Fold2 predicted conformations for our sample molecules and measured Flory radius from FLIM–FRET assays. A dot line indicates where the measured $R_F$ is equal to the Alpha Fold 2 predicted length. (**H**) Schematic diagram of protein conformational dynamics and two length scales, $R_F$ and Alpha Fold 2 prediction. Yellow and red stars indicate the two ends of the ectodomain.

The online version of this article includes the following figure supplement(s) for figure 4:

**Figure supplement 1.** Technical details for In situ FLIM–FRET measurements for protein sizes and conformational predictions.

in $\nu$ indicates that glycosylation increases the size of the protein at equilibrium, but the change in $R_F$ is slight, for example, a 1.3-fold increase for one of the longest ectodomains with N=4000 when these values of $\nu$ are applied. This calculation also gives a rough estimate of the upper limit of the $R_F$ of the extracellular domains of all membrane proteins in the human genome (approximately 10.5 nm). Physically, this change in $\nu$ by glycosylation may be caused by the increased intramolecular exclusion induced sterically between glycan chains. This estimated $\nu$ is much smaller than that of 0.6 for polymers in good solvents, possibly due to protein folding or anchoring effects on the membrane. In fact,

the $\nu$ of an intrinsically disordered protein in solution has been reported to be close to 0.6 (*Riback et al., 2019*; *Tesei et al., 2024*). Overall, these analyses using the Flory model provide information on the size distribution of membrane proteins and the influence of glycans, although the model cannot predict the exact size of each protein due to its specific folding.

We also analyzed protein conformation. Conformational data for the ectodomain of all 2515 membrane proteins in our list were obtained from the Alpha Fold 2 prediction (*Jumper et al., 2021*) and the end-to-end distances of the proteins in these conformations were plotted (*Figure 4F*, *Figure 4—figure supplement 1E-F*). Despite the large variation in this index among all proteins, the predicted Flory radii based on the measured scaling exponent $\nu$ were intermediate of all data plots, suggesting that protein sizes measured in our FLIM-FRET assays and the predicted conformations were in some agreement. Among the molecules we tested, the measured $R_F$ were consistent with or smaller than those calculated from the Alpha Fold predictions (*Figure 4G*). However, there were molecules, such as VCAM1 and EPHB1, for which the end-to-end distance in conformation was predicted to be much larger than the Flory radius estimated in our measurements (*Figure 4G*). There were many other molecules in the 2515 molecules that were plotted much higher than the predicted Flory radius, as shown in *Figure 4F*. Many of the molecules in this class are low glycosylated and tend to consist of multiple folded domains connected by flexible linkers, as seen in VCAM1 and EPHB1 (*Figure 4H*, *Figure 4—figure supplement 1I*). In contrast, the highly glycosylated proteins such as MUC1 (0TR) were predicted to be more globular and have fewer folded domains (*Figure 4H*, *Figure 4—figure supplement 1E-I*).

This discussion highlights the difference in size from $R_F$ measurements and predicted conformations: the $R_F$ is derived from an ensemble average, while Alpha Fold2 suggests only one representative conformation. Therefore, protein size estimates from highly anisotropic conformations may deviate significantly from the $R_F$. This is the case for low glycosylated proteins such as EPHB1 and VCAM1, which should have a very broad conformational landscape around its $R_F$ size (*Figure 4H*). This may suggest that protein glycosylation reduces the diversity of protein conformational space. This may be caused by the intramolecular interactions between amino acids and domain folding being interfered with by the glycans. If protein glycosylation functions to limit conformational dynamics, proteins may lose macroscopic flexibility with this modification. To test this hypothesis, further experimental validation or prediction with next-generation tools that include information on both conformational dynamics and multiresidue glycans would be needed (*Abramson et al., 2024*; *Tesei et al., 2024*).

## In vitro reconstitution of glycoprotein packing on membrane

The analysis in the previous section suggests that branching structures generated by molecular glycosylation may alter protein properties. Such changes may be important for the roles of glycans in infection inhibition (*Figure 4C*). In the context of polymer brush theory, such changes could modulate molecular packing. Then changes in molecular packing may alter $U^*$ associated with molecular exclusion, if that occurs in the formation of the interface (*Figure 3F*).

Under this hypothesis, we next investigated whether glycosylation affects protein packing and intermolecular interactions on the plasma membrane. To directly address this question, we biochemically reconstituted membrane protein systems in vitro and quantified protein packing. The ectodomain of the membrane proteins used in the viral infection assay was purified and introduced into fluid lipid bilayers formed on silica beads via affinity tag (*Figure 5A*). In this geometry, the membrane surface area was fixed and unchanged, allowing the molecular density to be measured without the influence of membrane structure (*Lee et al., 2021*). A non-glycosylated protein ectodomain produced in *E. coli* was also used for comparison. Molecular binding density was measured for each lipid-coated bead by flow cytometry.

The non-glycosylated recombinant CD43 and MUC1(14TR) ectodomain produced in *E. coli* bound at an estimated average saturation density of more than 10,000 molecules $\mu m^{-2}$, while the glycosylated CD43 and MUC1 ectodomain produced in HEK 293T cells bound at much lower saturation densities (*Figure 5B*). Thus, the saturation binding densities of these proteins reduced significantly by glycosylation. The ectodomain of EPHB1, a low-level glycosylated protein, bound in the intermediate density range. The binding density of non-glycosylated molecules was comparable to the saturation binding density shown for green fluorescent protein (GFP) on planar lipid bilayers (~7000 $\mu m^{-2}$; *Nye and Groves, 2008*). Dissociation constants for membrane-bound glycosylated (g-) proteins (0.61 μM

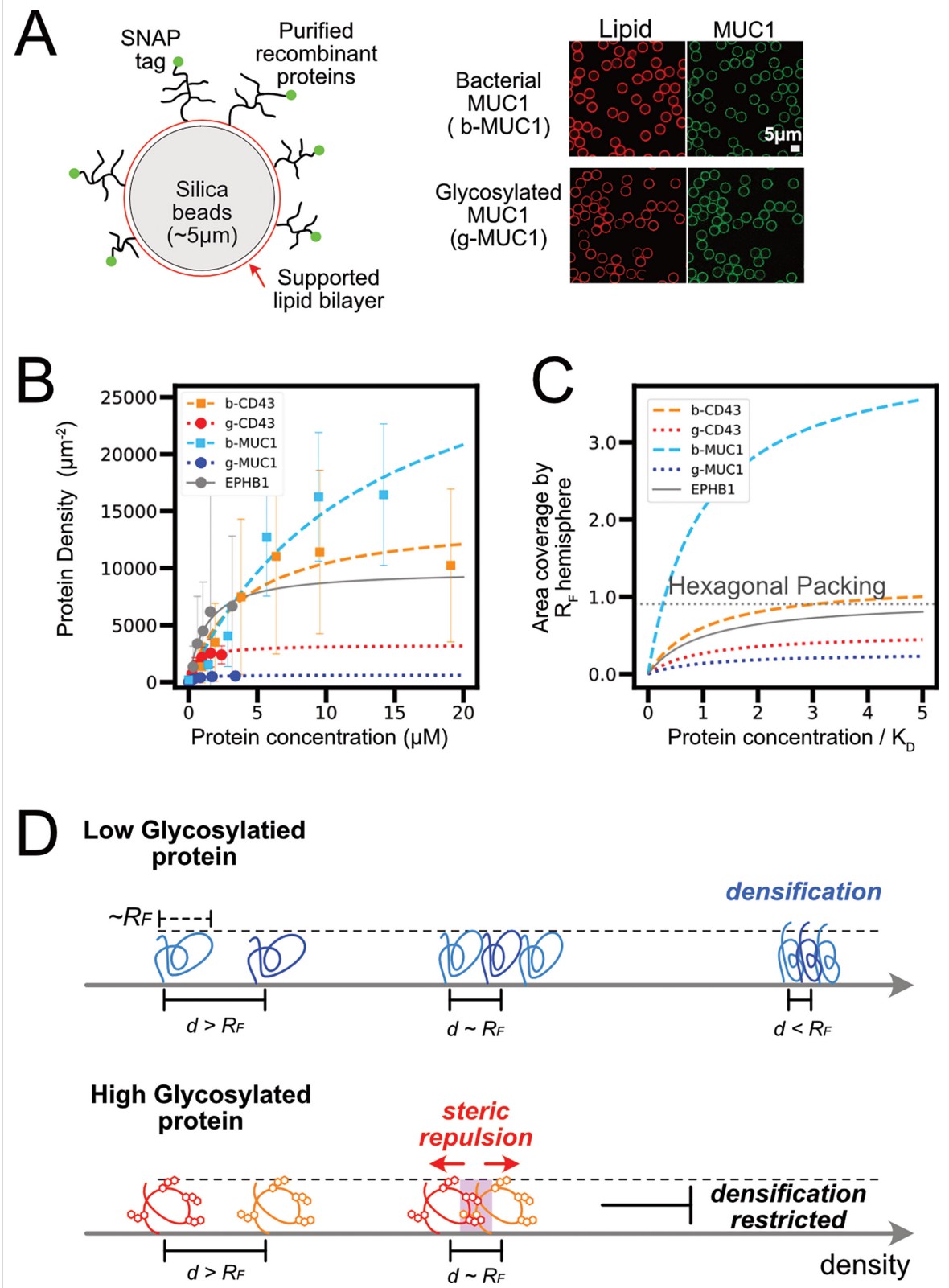

**Figure 5.** Biochemical reconstitution of protein packing in membrane surface. (**A**) Lipid bilayers coated on silica beads for incorporating bacterial (b-) and glycosylated (mammalian expressed, g-) proteins, schematic and fluorescent images. (**B**) Representative result for flow cytometry analyses of protein binding to lipid bilayer coated silica beads. Bar is the standard deviation in each measurement. Lines are regression curves to receptor binding model $Bx/(K_D + x)$, where x is protein concentration, $K_D$ is the dissociation constant, and B is the saturated density. (**C**) Relations of surface area coverage by

*Figure 5 continued on next page*

*Figure 5 continued*

bound proteins and concentrations of proteins used for membrane binding. Surface area was normalized by assuming all bound proteins were in a hemisphere of radius $R_F$ and the ratio of coverage was calculated. Protein concentrations were normalized by $K_D$. The dot line in the plot indicates the coverage when the hemisphere of radius $R_F$ aligned in a hexagonal close packing. (**D**) Schematic for structures and free energies for glycosylated and non-glycosylated proteins with similar $R_F$ that are at diluted and more condensed densities on the membrane surface.

The online version of this article includes the following figure supplement(s) for figure 5:

**Figure supplement 1.** Dynamics is not a factor to influence the packing of glycoprotein.

and 0.48 μM for CD43 and MUC1, respectively) were similar to or lower than those for bacterial (b-) proteins (3.84 μM and 12.6 μM for CD43 and MUC1, respectively), indicating that differences in molecular affinity for the lipid bilayer are not responsible for the large differences in saturation binding densities. The binding of all tested molecules was well explained by the standard first-order receptor-ligand binding reaction (*Figure 5B*), and single molecule tracking experiments confirmed that both glycosylated (g-) and non-glycosylated and bacterial (b-) proteins were equally diffusible as individual molecules (*Figure 5—figure supplement 1A*). Therefore, it is unlikely that there are protein-protein or protein-layer interactions that would allosterically affect protein packing.

To compare the packing of proteins with different molecular weights and $R_F$, we introduce the projected coverage area on the membrane as a normalized parameter, assuming that these molecules could be approximated by a hemisphere of $R_F$, when they are in the mushroom regime (*Halperin, 1999*; *Milner, 1991*; *Figure 5C*). Under this assumption, the coverage of highly glycosylated proteins at binding saturation was ~20% for g-MUC1 and ~40% for g-CD43, respectively. These were smaller than the coverage of molecules at hexagonal close packing that is ~90.7%. In contrast, the coverage of b-CD43 and b-MUC1 at saturated binding was estimated to be greater than 100% under this normalization standard, indicating that the mean projected sizes of these molecules in surface direction were smaller than those expected from their $R_F$. Thus, it is clear that glycosylation reduces the saturation density of membrane proteins, regardless of molecular size.

Highly glycosylated proteins resisted densification, indicating that some intermolecular repulsion is occurring. In the framework of polymer brush theory, the intermolecular repulsion of densely packed highly glycosylated proteins is due to an increase in either $f_{el}$, $f_{int}(d<R_F)$, or both (*Hansen et al., 2003*; *Wu et al., 2002*). The term of intermolecular interaction, $f_{int}$, is regulated by intermolecular steric repulsion, which occurs when neighboring molecules cannot approach the excluded volume created by the stochastic configuration of the polymer chain (*Attili et al., 2012*; *Faivre et al., 2018*; *Kreussling and Ullman, 1954*; *Kuo et al., 2018*; *Paturej et al., 2016*). The magnitude of this steric repulsion depends largely on $R_F$ in dilute solutions, but the molecular structure may also affect it when molecules are densified on a surface. In other words, the glycans protruding between molecules can cause steric inhibition between neighboring proteins (*Figure 5D*). Such intermolecular repulsion due to branched side chains occurs only when the molecules are in close proximity and sterically interact on a two-dimensional surface, but not in dilute solution, and does not occur in unbranched polymers such as underglycosylated proteins (*Figure 5D*). Based on the above, we propose the following model for membrane proteins: Only when the membrane proteins are glycosylated does strong steric repulsion occur between neighboring molecules during the densification process, suppressing densification.

The molecular structural state of these proteins needs to be further discussed to estimate the contribution of $f_{el}$, which represents resistance to molecular elongation. Our results suggest that these densely packed nonglycosylated molecules are no longer in a free mushroom state. However, their saturation density was several times lower than previously reported brush transition densities, such as 65,000 μm$^{-2}$ for 17 *kDa* polyacrylamide ($R_F$ ~15 nm) on a solid surface (*Wu et al., 2002*). To compare our data on fluid bilayers with previously reported data on solid surfaces, we performed additional experiments with lipid bilayers that lost fluidity. No significant changes in protein binding between fluid and nonfluid bilayers were observed for both b-MUC1 and g-MUC1 molecules (*Figure 5—figure supplement 1B*). This result suggests that membrane fluidity does not affect the average intermolecular distance or other relevant parameters that control molecular binding in the reconstituted system. Based on these, we speculate that the saturated protein density observed in our experiments is lower than or at most comparable to the actual brush transition density. Thus, although these crowded proteins may be restricted from free random motion, they are not significantly extended as in the

condensed brush state, in which the contribution of resistance to molecular extension $f_{el}$ is expected to be small relative to the overall free energy of the system.

Note that this does not mean that glycoproteins cannot form condensed brush structures: in fact, highly glycosylated molecules (e.g. MUC1) can form brush structures in cells when such proteins are expressed at very high densities (*Shurer et al., 2019*). In these cells, plasma membranes were forced to accommodate these large amounts of membrane proteins with hydrophobic transmembrane domains. In contrast, in our experiment, proteins were soluble in the aqueous phase, and we can analyze intermolecular repulsion in ectodomain independently from the effect of protein compartmentalization. In addition, cells containing brushed glycoproteins also transformed the cell membrane into a shape with an enormous amount of tubular structure. Such membrane deformation results in the increase of total surface area to reduce the density of glycoproteins, indicating that there is strong intermolecular repulsion between glycoproteins. In any case, the free energy of the system is determined by the balance between protein binding and insertion into the membrane, protein deformation, and repulsive forces between proteins, which determine the density of proteins depending on the configuration of the system. Thus, although strong intermolecular repulsions were prominently observed in our simplified system, this may not be the case in other systems. We also note that it has been previously shown that membrane tubules generated upon the insertion of polymers in membranes were neither inhibitory nor activating on viral infection (*Kaizuka and Machida, 2023*). This is simply because these tubular structures are on the ~μm scale, and much larger than viruses, and we assumed that this logic can also be applied to membrane tubules generated by overcrowded glycocalyx (*Shurer et al., 2019*).

## Super-resolution imaging of virus-cell interface

Glycosylation of proteins induces intermolecular repulsion, which affects the distribution of molecules on the membrane. Here, we measure the distribution of membrane proteins in situ to determine whether protein glycosylation induces molecular exclusion or trap at the virus-cell interface (*Figure 3D*). To address this question, we used dual-color super-resolution microscopy (STORM) imaging to locate each membrane glycoprotein and viral Spike proteins (*Rust et al., 2006*). We found that the distribution patterns of these proteins are very diverse (*Figure 6A*). To analyze these diverse results in an unbiased manner, we introduced a cross-correlation function $C(r)$ (*Figure 6B–C*). $C(r)$ here is the probability distribution of distances between cellular and viral proteins, calculated as a radial normalization of the histogram of the mutual spacing distances $r$ between cellular and viral proteins in the image (*Figure 6B*; *Schnitzbauer et al., 2018*; *Stone and Veatch, 2015*; *Veatch et al., 2012*). Although the cross-correlation function has often been used to analyze molecular co-localization in super-resolution images (*Schnitzbauer et al., 2018*; *Stone and Veatch, 2015*; *Veatch et al., 2012*), we intended to apply this function over a longer distance range to analyze protein distributions at the virus-cell interface. Since the localization of Spikes was limited in viruses that were only sparsely distributed throughout the image, $C(r)$ in our study can somewhat reflect the average probability distribution of cellular proteins at distance $r$ from these viruses in each image. In most of the $C(r)$ plots, a peak appears at 0~100 nm, which is near the virus diameter, and the function decays as $r$ increases (*Figure 6C*). This trend may be attributed to the fact that the Spike as well as the cellular proteins were distributed in specific locations (e.g. discrete viruses and plasma membrane), and the rest of the region in images was mostly blank (*Figure 6A*). In contrast, $C(r)$ for molecules uniformly distributed throughout the image tends to converge to 1 as $r$ increases (*Schnitzbauer et al., 2018*; *Stone and Veatch, 2015*; *Veatch et al., 2012*).

The next step is to obtain a measure of the protein density at the virus-cell interface from $C(r)$. To solve this inverse problem, the information of relationship between $C(r)$ and the interface molecular density is needed. To obtain this relationship, we adopted the approach of using simulated images. We prepared a set of $C(r)$ plots calculated from simulated images of various types of virus-cell membrane interfaces (*Figure 6D*). These images were characterized by the parameter *density_(in/out)*, which is the ratio of the mean density of membrane proteins inside and outside the virus-cell interface. Thus, this parameter *density_(in/out)* is the measure to know whether proteins at the virus-cell interface are excluded or not.

Simulated images included those in which the virus is isolated from the cell membrane ('separation') and those in which cell membrane molecules are concentrated on the virus and *density_(in/*

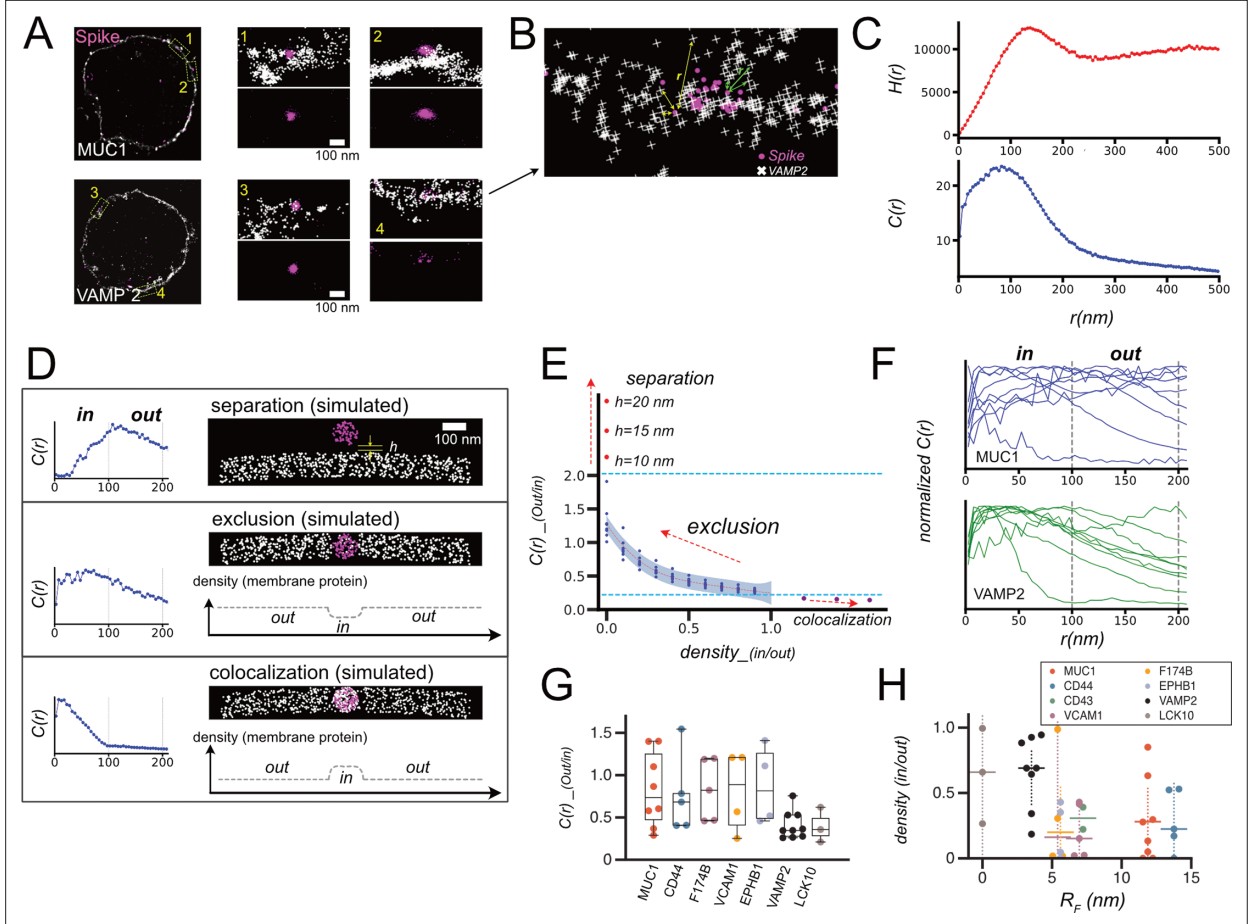

**Figure 6.** Superresolution imaging of virus and cellular proteins for analyses of virus – cell interface. (**A**) Dual color STORM images of SNAP-MUC1/ SNAP-VAMP2 in cells and Spike in SARS-CoV-2-PP bound to the cells. Whole cell images (left) were reconstructed from STORM data, and coordinates determined by STORM were individually plotted in expanded images on the right. (**B**) Schematic of calculation of cross-correlation. Mutual distance between all combinations of two dyes was calculated from the coordinates determined by STORM imaging. (**C**) Examples of histogram of mutual distances between all combinations of two dyes were calculated from coordinates H(r), and the cross-correlation function C(r) that is the normalized radial average of H(r). Δr, a size of shell and the bin size for histogram, was set to be 5 nm. (**D**) Examples of simulated STORM images, and calculated C(r) from these images. h denotes the distance between virus and cell when they are separated, and density plot illustrates average protein densities along membranes in these images. (**E**) Plot of C(r)_(out/in) calculated for all simulated images, along with density_(in/out) and h. Blue dashed lines indicate upper and lower bounds for C(r)_(out/in) calculated for stable virus – cell interface, and red dot line is the fifth polynomial regression to data point and purple area represents one sigma below and above the line. (**F**) Traces of C(r) for individual cells expressing MUC1 and VAMP2. (**G**) C(r)_(out/in) calculated for all STORM images. (**H**) density_(in/out) for all STORM images converted from C(r)_(out/in) based on the regression in E was plotted along with $R_F$ for each protein determined by FLIM – FRET (**Figure 4**).

The online version of this article includes the following figure supplement(s) for figure 6:

**Figure supplement 1.** STORM images of all analyzed cells, expressing designated proteins.

out)>1 ('co-localized'). To organize the list of calculated C(r) for all simulated images, we introduced the index C(r)_(out/in), which is the ratio of the integral of C(r) at r=100–200 nm to the integral of C(r) at r=0–100 nm. This boundary r~100 nm was set since the size of virus is approximately 100 nm, and we assumed that this index reflects the ratio of the probability distribution of proteins inside and outside the viral–cell interface. Indeed, we found that C(r)_(out/in) increased continuously from ~0.2 to~2.0 at protein exclusion interfaces where density_(in/out) ranged from 0 to 1 and the densities were in the range of those in typical STORM images. In addition, the C(r)_(out/in) for 'separation' and 'co-localization' were found to be >2.0 and<0.2, respectively (**Figure 6E**). Altogether, among these three types of cell–virus interface (co-localization, exclusion, and separation), the parameter C(r)_(out/in) increases monotonically along with the value of density_(in/out). Therefore, the average density_(in/out) for each image can be calibrated from C(r) using this relationship.

Next, images of cells with *C(r)_(out/in)<2.0*, which are not in 'separation' and represent about 80% of all cells, were analyzed. As indicated above, the majority of viruses in these images would be expected to form an interface with the cells. *C(r)_(out/in)* and calibrated *density_(in/out)* were compared between molecules. *Density_(in/out)* was found to be similarly low for groups of membrane proteins, and molecules with molecular weights greater than 50 kDa (MUC1, CD44, F174B, VCAM1, EPHB1, all tagged with SNAP) were similarly excluded at the virus-cell interface to an average ~80% reduction (*Figure 6GH*). In contrast, much higher *density_(in/out)* was observed for VAMP2, a SNAP-tagged-only control molecule, and LCK10, a control molecule without any ectodomain, respectively, suggesting that only very low amounts of both of these molecules were excluded. The slight decrease in *density_(in/out)* for these control molecules may be an artifact caused by the oversimplification in the simulation of cell images.

These results showed that membrane proteins above a certain threshold size are excluded from the virus-cell interface, regardless of the degree of glycosylation. Interestingly, the transition from low-exclusion regime to high-exclusion regime occurred nonlinearly and abruptly with protein size (*Figure 6H*). This may be due to measurement bias in part, since we analyzed only viruses bound to the cell membrane but not ones already internalized in the cell. More viruses may be internalized in cells with smaller proteins, but they are not included in the analysis, and thus the rate of protein exclusion may be overestimated for cells with these smaller proteins. Nevertheless, the overall correlation between Flory radius and *density_(in/out)* can be calculated to be –0.36.

## Discussion

We hypothesize that the energy barrier *U\** required for the virus to form an interface with the cell is increased by glycosylation of membrane proteins in the plasma membrane, thereby suppressing infection. We summarize the mechanism based on our analyses. In the early stages of interface formation, the receptor and ligand form the binding point at the center, and there is a region with a longer interfacial distance around this center (*Figure 7A*). As we have already seen, membrane proteins above a certain size are excluded from this region and an interface is formed (*Figure 6H*). The driving force for this molecular exclusion may be the adhesion energy of the interfacial membrane. Since membrane adhesion at the interface is a process that stabilizes the energy level of the system, molecular exclusion should be promoted. In reality, however, there are many viruses that bind to cells but are not taken up due to insufficient molecular exclusion at the interface (*Figure 3A*, *Figure 6—figure supplement 1*). This means that there is an energy barrier that resists molecular exclusion from the interface.

A possible factor creating an energy barrier to molecular exclusion is the strong viscosity of the plasma membrane, experimentally confirmed to be on the scale of ~1 $\mu m^2/s$ (*Figure 5—figure supplement 1A*). In such an environment of strong viscosity, when molecular exclusion from the interface bond center to the periphery proceeds, viscous stress is generated to resist the movement (*Figure 7A*). Viscous stress, however, is created in the lipid bilayer and is common to all transmembrane molecules, regardless of protein glycosylation.

On the other hand, the glycosylation of ectodomain affects the intermolecular repulsion on the plasma membrane when the molecular density increases (*Figure 5D*). Are there situations in which this repulsion has an effect on molecular exclusion? We assume that the density of excluded molecules increases at the outer edge of the interface and that intermolecular interaction there is strongly affected by glycosylation (*Figure 7B*). Molecular exclusion occurs only in the region inside the viral interface, while molecules outside the interface only diffuse. And the cell membrane is highly viscous. As a result, we speculate that molecules may become locally and instantaneously crowded in the vicinity of the interface. In such a geometry, the interface between the virus and the cell membrane can only be formed if the virus and membrane proteins are all squeezed into that narrow area (*Figure 7C*). If the density increase is sufficiently large, the glycosylated molecules may experience strong intermolecular repulsion, preventing them from being packed together with the virus. On the other hand, molecules with low glycosylation may be readily distorted and accept crowding by being brushed. We propose that this difference creates an energy barrier that would kinetically inhibit the processes of interface formation and molecular exclusion.

Based on this model, we can classify the structure of the interface into three types depending on the type of cell membrane protein. The first is the case where the membrane protein is small and thus infection can proceed even if the protein is trapped at the viral cell membrane interface, at least in

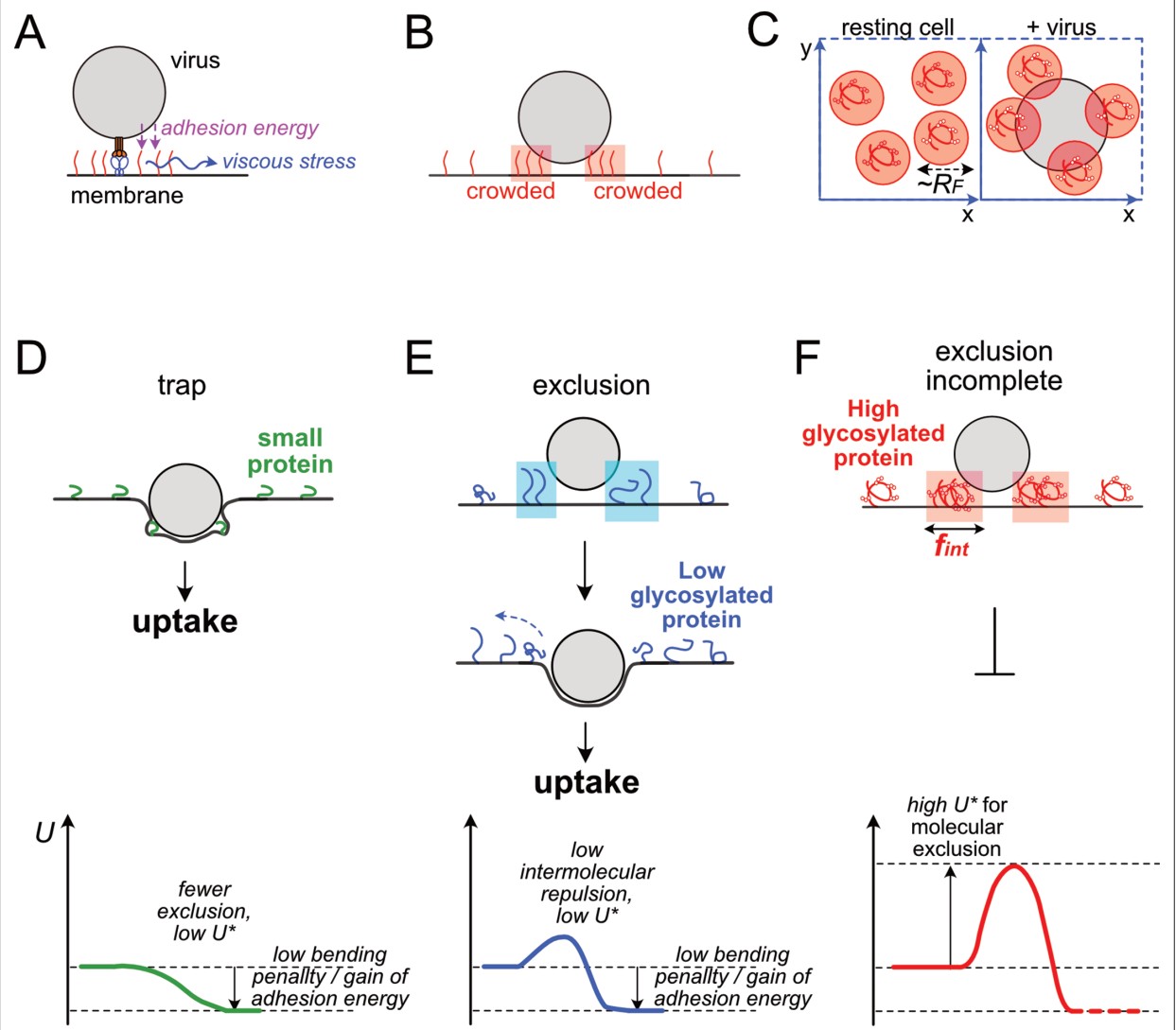

**Figure 7.** Models for three distinct cases of virus – cell interface. (**A**) Energy and force involved in molecular exclusion from virus-cell interface. (B) Transient distribution of membrane proteins nearby the interface. (C) Two-dimensional distribution of viruses and glycoproteins on the surface horizontal direction of the cell membrane. (D-F) Schematic of virus-cell interface structure and corresponding free energy chart. (D) Virus-cell interface with very small membrane proteins. Due to the low energy penalty in constructing virus-cell interface entrapping these small molecules, the virus can form the interface without excluding proteins and can infect cells. (E-F). Virus-cell interface structure for larger membrane proteins. Interface formation requires protein exclusion. Due to membrane viscosity and inhomogeneous field in adhesive energy, excluded proteins become crowded at proximity of the interface. In the case for highly glycosylated proteins (**F**), intermolecular repulsion between excluded proteins becomes very high at high density regions. Such repulsion generates a high kinetic barrier for molecular exclusion, preventing the process from proceeding to subsequent steps for infection. In contrast, in the case of low glycosylated proteins (**E**), their intermolecular repulsion and energy barrier are lower, and thus these molecules are easier to be excluded from the interface, and virus infection is not strongly inhibited.

low-density situations (*Figure 7D*). For highly glycosylated molecules above a certain size, the glycans interfere with molecular exclusion by generating strong intermolecular repulsion and preventing successful interface formation. As a result, viral infection is inhibited (*Figure 7F*). For molecules above a certain size but with low glycosylation, intermolecular repulsion is weak and molecular exclusion is easily enabled, resulting in the formation of an interface and progression to infection (*Figure 7E*).

This nonequilibrium model may be mathematically represented by incorporating terms for viscous stress and molecular diffusion. If the entire process can be mathematically modeled, it may be possible to relate the energy barrier $U^*$ to the average molecular density and $R_F$. If such a model can be successfully incorporated into a hierarchical Bayesian model, it may be possible to relate the IC50

value of each molecule to the number of glycans. This model may explain the sharp decline in infection efficiency that could not be reproduced by the simple Sigmoid model (*Figure 1F*).

When the virus receptor is a glycoprotein or glycan itself, the inhibition of virus infection by glycans becomes more complex because the total amount of glycans is also a function of the receptor density. It is also important to note that the total amount of infection into a cell is the time integral of virus entry. Even if the probability of virus entry is significantly reduced by glycans, the cumulative number of virus entries may increase if high concentrations of virus continue to be supplied from outside the cell for a long period of time. In the case of Adenovirus, which continues to amplify in HEK293T cells after infection, we showed that MUC1 on the cell surface has an inhibitory effect on long-term cumulative infection (*Figure 1—figure supplement 1F*). However, such an accumulation effect may be case-by-case depending on the virus cell system and may be more pronounced when the cell surface density of virus receptor molecules is high. As a result, if the virus receptor molecule is a glycan or glycoprotein and infection continues for a long period of time, the infection inhibition effect may not be observed despite an apparent increase in the total amount of glycans in the cell. In any case, our results clarified the factor of virus entry inhibition dependent on the total amount of glycans because appropriate conditions were set.

In this study, we attempted to generalize the surface structure on the cell side, but the surface structure on the virus side may also have an effect. The efficiency of virus adsorption and the efficiency of cell membrane protein exclusion from the interface will change depending on the molecular length of the receptor-ligand, although receptor priming also has an effect. In addition, free ligands of the viral envelope or other coexisting glycoproteins may also have an effect as they are also required for exclusion from the virus-cell interface. In fact, there are reports that expression of CD43 and PSGL-1 on the virus surface reduces virus infection efficiency (*Murakami et al., 2020*). Such interface structure may be one of the factors that determine the infection efficiency that differs depending on the virus strain. More generally, modification of the surface structure may be effective for designing materials such as lipid nanoparticles that construct the interface with cells.

This paper has laid the foundation for a strategy to describe molecular nonspecific functions by means of molecules. The next step, for example, would be to obtain quantitative relationships between the cell transcriptome and the amount of glycans on the cell surface, where machine learning may be useful. Future goals of these kinds of studies would be to treat nonspecific phenomena within the same framework as other molecule- and gene-specific functions. In addition, even with the current advances in machine learning and single-cell data technology, it is still difficult in many areas to predict cellular-level phenotypes from genomics and transcriptomes. Therefore, we believe it is useful to mix such auxiliary methods that utilize a combination of statistics and rule-based physical models.

## Methods
### Cell lines and materials
All reagents are listed in the *Supplementary file 1*, with information of software used in this study. HEK293T and Calu-3 cells were obtained from ATCC and had been authenticated by the vendor. All cell lines were routinely tested for mycoplasma contamination by the Nanobio core facility in the National Institute for Materials Science.

### List of membrane glycoproteins in human genome
From UniProt database (*Homo sapiens* UniprotKB, 2021 September release), amino acid sequences of all membrane-associated proteins, either topologically containing domains associated with 'extracellular' or GPI-anchored, with positive in glycosylation either experimentally or by prediction were collected. These sequences were then used as input to two software packages, NetNGlyc ver 1.0/NetOGlyc ver 4.0 (https://services.healthtech.dtu.dk/service.php?NetNGlyc-1.0, https://services.healthtech.dtu.dk/service.php?NetOGlyc-4.0), and GlycoEP (https://webs.iiitd.edu.in/raghava/glycoep/submit.html). Glycosylation sites were predicted for all these sequences. Default settings in both software packages were used and the setting of Binary profile of patterns was selected for GlycoEP. Then, the candidate sites were selected by thresholding in scores, based on the software recommendations. To meet the 4000 amino acid sequence maximum cutoff in NetNGlyc, part of cytoplasmic sequences in three long proteins (UniProt Entry: Q14517, Q07954, Q9NYQ8) were eliminated

in the calculations. Next, we identified the ectodomain regions of these proteins by collecting topological information from UniProt. For multiple transmembrane proteins, only the largest extracellular domain was selected. Then, candidate glycosylation sites only between these ectodomain regions were selected, and compiled and counted for making the final list (*Supplementary file 2* and *Supplementary file 3*). For convenience of availability, mouse genes for Dll4 and Jag1 were used in the infection inhibition assay, and the predicted glycosylation of these genes was calculated separately (glycosylation rate/AA was 0.0837 and 0.0584 for the human DLL4 gene and mouse Dll4 gene, respectively, and 0.0513 and 0.0334 for the human JAG1 gene and mouse Jag1 genes were 0.0513 and 0.0334, respectively). Results are compiled in the *Supplementary file 2* and *Supplementary file 3*.

## PNA binding assay and reaction model

HEK293T cells were transiently transfected with various membrane proteins 24 hr prior to the measurement. These cells that were pre-stained with 5 µM SNAP-Surface 647 (New England Biolabs) for 1 hr at room temperature, if necessary, were stained with 5 µg/mL of Alexa488-labeled lectins (PNA, WGA (Wako Fujifilm) or SSA (J-Chemical)) for 1 hr at room temperature. These labeled lectins were obtained by labeling in-house with Alexa488 NHS ester (Thermo Fisher). Ligands with differential degrees of labeling were prepared in order to have signals in a wide range without signal saturations in flow cytometry assay. After washout with PBS, cells were subject to flow cytometry assay, using SH800 (Sony).

The binding of PNA lectins to cells was modeled by making various modifications to the standard ligand-receptor binding. Each sample consists of a test tube of cells expressing one kind of membrane protein. Reaction for each sample is batch-based, with excess PNA ligand of concentration [L] in a tube. The reaction was approximated to be a first-order binding with the receptor, which is an individual sugar chain on a membrane protein. Due to steric hindrance, it was assumed that at most only one lectin binds to one sugar chain. It was also assumed that PNA binding reaction is governed by the same kinetics for glycans branched from all the different membrane proteins. In other words, all glycans of all tubes are equivalent in the reaction kinetic. Glycans in all tubes are assumed to be in equilibrium in the binding to lectin, as [L] is very excess, despite the variance of total number of glycans between tubes. Thus, the only difference between the samples is the number of glycans per one protein molecule. In this setting, PNA binding to the sugar chains at equilibrium is described as follows

$$ K_D = \frac{[L]\,[R]}{[LR]} $$

where $K_D$ is the dissociation constant, and [R] and [LR] are the unbound and ligand-bound receptor sugar chains of each cell, respectively. Assuming that the diameter and surface area of all cells are equal, [R] and [LR] in this equation can be replaced by the total number of ligand-unbound receptor sugar chains in the cell. On the other hand, when $G_i$, the number of glycosylation sites that are specific to protein type i, is introduced, the total number of glycans in the cell [R_total] should be described as follows:

$$ [R\_total] = [P] * G_i, $$

where [P] is the number of proteins in the cell. [R_total] is expressed as

$$ [R\_total] = [R] + [LR] , $$

where [R_total] is different in all cells. For first-order reactions, a linear relationship is expected between [LR] and [R_total], which can also be translated into a relationship with [P]:

$$ \frac{[LR]}{[R\_total]} = \frac{[LR]}{[P] * Gi} $$

where [P] and [LR] are measured as signals from the fluorescent SNAP substrate and PNA, respectively, in the flow cytometry assay. If the flow cytometry data for cells expressing protein type i can be regressed to a linear relationship, the slope [PNA/mol]_i should be

$$\left[\frac{PNA}{mol}\right]_i = [LR]/[P] = [LR]/[R\_total] * Gi = \frac{[LR]}{([R] + [LR])} * Gi = \frac{1}{(KD/[L] + 1)} * Gi.$$

Since KD and [L] were constant for all proteins, the [PNA/mol]$_i$ obtained from the linear regression for each protein i should have a linear relationship with G$_i$ across all different proteins.

## Virus production

All viruses were produced in HEK 293T cells as previously described (*Kaizuka and Machida, 2023*). SARS-CoV-2 pseudoparticles were produced using pLV-eGFP (obtained from Pantelis Tsoulfas), pCAG-HIVgp (obtained from Hiroyuki Miyoshi, Riken BRC, Japan), and a vector where VSV-G gene in the pCMV-VSV-G-RSV-Rev (obtained from Hiroyuki Miyoshi, Riken BRC, Japan) was replaced with Spike gene (SARS-CoV-2, Wuhan-Hu-1, obtained from SinoBio, China). Lentiviruses were produced using pLV-eGFP, pCAG-HIVgp, and pCMV-VSV-G-RSV-Rev. Adeno-associated viruses were produced using plasmids of AAV GFP, pAdDeltaF6, and pAAV2/2 (obtained from Fred Gage, James M. Wilson, and Melina Fan, respectively). Adenoviruses were produced by replicating AxCAEGFP, generated from backbone vector pAxcw that contains genes of AdV511,12 (obtained from Dr. Murata, Riken BRC). Viruses were collected from supernatant medium, and if necessary, concentrated using Lenti-X concentrator kit (Takara bio, Japan). Lenti-X p24 Rapid Titer Kit (Takara Bio) was used to measure the titer of collected SARS-CoV-2-PP by ELISA (Enzyme-Linked Immunosorbent Assay). For virus binding assay, SARS-CoV2-PP was prepared by labeling with 0.4 µM DiIC18(5) solid (1,1'-Dioctadecyl-3,3,3',3'-Tetramethylindodicarbocyanine, 4-Chlorobenzenesulfonate Salt, or DID) (Thermo Fisher) with Biospin6 (Bio-Rad) to remove residual dyes. Binding of viruses was measured by flow cytometry after washing cells.

## Virus binding and infection assay and measurement of cell protein density

Cells were cultured in standard conditions using medium (DMEM for HEK 293T cell, and EMEM for Calu-3 cell) supplemented with 10% FBS. Air-liquid interface culture of Calu-3 cells was conducted using Transwell (Corning, USA), according to the protocol provided by the manufacturer.

HEK293T cells transiently transfected with ACE2-iRFP, TMPRSS2-TagRFP657, and target membrane protein tagged with either mTagBFP2 or SNAP at 24 hr prior to the virus infection. These cells were pre-stained with 5 µM SNAP-Surface 488 for 1 hr at room temperature if necessary and were analyzed by flow cytometry to quantify the surface densities of ACE2 and the target membrane protein. These cells from the same batch were then infected with SARS-CoV2-PP or other viruses for 3 hr at 37 °C in suspension. Viruses were administrated at ~0.015 pg/cell according to the p24 ELISA for virus stock solutions, and typically about 20% of cells were infected in the absence of any inhibitory protein expression. After removal of viruses, infected cells were cultured for 2 days. Then, cells were collected for flow cytometry assay to analyze infection by measuring the expression of GFP gene introduced by infected viruses. In the virus binding assay, cells were measured by flow cytometry after 3 hr of incubation of cells with viruses that were stained with DID.

Membrane proteins and functional tags used in this study were SNAP-CD24, SNAP-CD43, SNAP-CD44, SNAP-164, SNAP-SDC1, SNAP-F174B, SNAP-VCAM1, SNAP-VAMP2, SNAP-PD-1, SNAP-MUC1 (42TR), SNAP-MUC1 (0TR, deletion in aa 32–961), SNAP-MUC1 (4TR, deletion in aa 32–879), SNAP-MUC1 (14TR, deletion in aa 32–703), PODXL-mTagBFP2, TMEM123-mTagBFP2, EFNB2-mTagBFP2, EPHB1-mTagBFP2, Jag1-mTagBFP2, Dll4-mTagBFP2, and GYPC-mTagBFP2. They were all constructed in the pAcGFP-n1 vector (Invitrogen) where AcGFP was either removed for SNAP tagged construct or replaced with mTagBFP2.

Protein surface densities as well as GFP expression were calibrated from flow cytometry data, in similar manner as previously described (*Kaizuka et al., 2021*; *Kaizuka and Machida, 2023*). This calibration method is based on the calibration strategy between intensities of two different dyes using fluorimeter, developed for microscopy imaging (*Galush et al., 2008*). Liposome standards containing fluorescent molecules (0.01–0.75 mol% perylene [Sigma], 0.1–1.25 mol% Bodipy FL [Thermo Fisher], and 0.005–0.1% DiD) as well as DOPC (Avanti polar lipids) were measured in flow cytometry (*Figure 1—figure supplement 1L*). Meanwhile, by fluorimeter, fluorescence signals of these liposomes and known concentrations of recombinant mTagBFP2, AcGFP, and TagRFP-657

proteins and SNAP-Surface 488 and Alexa 647 dyes (New England Biolabs) were measured in the same excitation and emission ranges as in flow cytometry assays (*Figure 1—figure supplement 1M*). Ratios between the integral of fluorescent intensities in this range between two dyes of interest are used for converting the signals measured in flow cytometry. Additional information needed for calibration is the size difference between liposomes and cells. The average diameter of liposomes is measured to be 130 nm, and the diameter of HEK 293T cells is estimated to be 13 μm (*Furlan et al., 2014*; *Ushiyama et al., 2015*). From these data, the signal from cells acquired by flow cytometry can be calibrated to molecular surface density. For example, the Alexa 647 signal acquired by flow cytometry can be converted to the signal of the same concentration of DID dye using fluorometer data, but the density of the dye is unknown at this point. This converted DID signal can then be calibrated to the density on liposomes rather than cells using liposome flow cytometry data. Finally, adjusted for the size difference between liposomes and cells, the surface molecular density on cells is determined. By going through one cycle of these procedures, we could obtain a calibration unit, such as 1 flow cytometry signal for a cell in the designated illumination and detection setting = 0.0272 mTagBFP2 μm$^{-2}$ on cell surface. All fluorescent proteins were assumed to be matured and fluorescent. For SNAP-tagged proteins, we confirmed that the binding of fluorescent substrate to cells expressing SNAP-tagged membrane proteins was saturated when incubated with 5 μM substrate, and background fluorescent signal of nonspecific binding was minimal (*Figure 1—figure supplement 1G*). Thus, we also assumed that most of the SNAP-tagged proteins in cells were fluorescent and all the fluorescent signals from these proteins were detected in our assays. ACE2-TagBFP expressions varied between samples, which affects the infection rate (*Figure 1—figure supplement 1H*). We checked the ACE density of cells at the time of virus infection, and only cells with ACE2 expressed in the range where infection linearly increases with ACE2 density were chosen for infection assays, and the final results of GFP intensities were adjusted with the mean ACE2 densities measured at the time of infection and the calibration line (*Figure 1—figure supplement 1H*). Inhibition of infection was evaluated by the average density of target membrane proteins measured at the time of infection, and the infection rate of samples was normalized by the infection rate of control infection samples that did not express inhibitory proteins in the cells.

For the imaging of virus infections in air-liquid interface culture, after the 10 days of air-liquid interface culturing of Calu-3 cell monolayer in Transwell, SARS-CoV2-PP was administrated at ~0.015 pg/cell for 3 hr at 37 C, then cells were cultured for 2 days after the washout of residual viruses. Prior to imaging, cells were stained with Hoechst 33342, Alexa647 labeled lectins, primary and secondary antibodies, sequentially. Stacked images in the z axis were maximum projected by Fiji/ImageJ to compare signals in different channels. Pearson correlation and TOS correlation were calculated for maximum projection images and were performed by EzColocalization plugins in Fiji/ImageJ for all pixels without selecting any ROIs for cells (*Stauffer et al., 2018*). Thus, all pixels were classified depending on the intensity of two signals in unbiased manners. Then, expectations for two signals were calculated separately in all these classes, resulting in a computed TOS matrix in which those numbers indicate whether the number of objects classified in each region is higher or lower than expected for uniformly distributed data.

## Endocytosis assay

1.0 mg/mL FITC-labeled 10 kDa dextran (Tokyo Chemical Industry Co., Ltd.) was incubated with HEK293T cells expressing various membrane proteins at 37 °C for 30 min. After washing the cells twice with PBS, the FITC signal was measured by flow cytometry (SH800, Sony).

## Virus infection inhibition, hierarchical Bayesian modeling

Virus infection was assumed to be inhibited dependent on the density of inhibitor glycoprotein on cell membrane, and a standard sigmoidal dose-response curve for inhibitory reaction was introduced:

$$Y = \frac{1}{\left(1 + \left(X/\sigma_{IC50}\right)^n\right)}$$

where Y is the infection rate, X is the density of glycoprotein, and n is the Hill coefficient describing a variable from the ideal sigmoidal. Infection rate Y reaches 0.5 at the density = $\sigma_{IC50}$.

In hierarchical Bayesian modeling, for the Hill coefficient, a common prior distribution (normal distribution) was set for all molecules, and for the IC50 values for each kind of protein_i, a Half Cauchy prior distribution was set for each molecule separately because it is non-negative and can take a wide range of values (*Figure 1—figure supplement 1I*). In this setting with Gaussian noise, Markov chain Monte Carlo sampling was executed in the scheme of probabilistic programming language NumPyro, repeated for four times (https://num.pyro.ai). Once convergence was confirmed over four rounds of sampling, the common Hill coefficient and protein-specific IC50 were determined from the posterior distributions (*Figure 1—figure supplement 1I*).

## FLIM-FRET

FRET between a donor molecule of fluorescently labeled SNAP tag fused to the end of the ecto-domain of membrane protein and an acceptor molecule of small and nonspecific plasma membrane dye was measured. Live HEK293T cells expressing various SNAP-tagged membrane proteins were stained with 5 μM or less of SNAP-Surface 488 substrate, a modified ATTO 488 molecule, and with either membrane dyes, PlasMem Bright Red (Dojindo) or MemGlow 590 (Cytoskeleton). Doubly stained cells were imaged by Leica SP8 equipped with FLIM module, and the lifetime of fluorescence of donor ATTO 488 was measured and imaged in the FLIM mode. The lifetime was estimated as the time constant in the exponential decay in the histogram of time interval between the excitation laser pulse and emitted photons. A histogram was generated from ROI images cropped from cell membranes for each image, thus the measured lifetime reflected photon emissions from all dyes located in the image ROIs.

We introduced a simple model to analyze the height of protein ectodomain from the membrane, from FLIM-FRET data. In this case, the donor ATTO 488 dyes conjugated to the terminus SNAP tags were assumed to locate along the concentric circle at the fixed distance $z$ from the plasma membrane. Then, we also approximated that FRET occurred between a single donor and multiple acceptor molecules located in a nearby plane at the distance of $z$. This planar geometry approximation was made since the scale of Forster distance (~nm) is much shorter than cell diameter (~10 μm). FRET for a single donor to populations of acceptors in a plane can be obtained by radially integrating energy transfer between a single pair of donor and acceptor, $k_i = (1/\tau_D)(Ro/R)^6$, toward all acceptors in a nearby plane as (*Gibson and Loew, 1979*; *Wong and Groves, 2002*):

$$k_T = \frac{\sigma \pi R_0^6}{2\tau_D z^4}$$

where $Ro$ is Forster distance for the donor-acceptor pair, $R$ is actual distance between the two, $\tau_D$ is the lifetime of donor without FRET to acceptor, and σ is the density of acceptor. Then, the FRET efficiency E can be described as (*Gibson and Loew, 1979*; *Wong and Groves, 2002*):

$$E\_{model} = \frac{k_T}{k_T + \dfrac{1}{\tau_D}} = \frac{\sigma}{\sigma + \dfrac{2z^4}{\pi}\dfrac{1}{R_0^6}}$$

In FLIM measurement, FRET efficiency was measured by donor lifetime as follows: $E\_{FLIM} = 1 - \frac{F_{DA}}{F_D} = 1 - \frac{\tau_{DA}}{\tau_D}$

where $\tau_{DA}$ is the lifetime of donor with FRET to acceptor.

In our measurements, we imaged the lifetime of donors $\tau_{DA}$ in cell populations in several different mean acceptor densities. Therefore, we plotted $E\_{FLIM}$ for different σ, regressed the plot to $E\_{model}$, and obtained z as the distance from membrane to SNAP for each protein.

Forster distance Ro is specific for each donor-acceptor dye pair. In our experiments, we found that SNAP – surface 488 (Atto 488 dye) and PlasMem Bright Red were good FRET pairs, in the context of both FRET efficiency and dye stability in membrane localization. Unfortunately, details of molecular and optical information regarding PlasMem Bright Red are not disclosed, and Ro for this pair of dyes cannot be directly calculated. Therefore, we estimated the Ro for PlasMem Bright Red by comparing two experiments in the same setting with this dye and the other dye with known Ro value, Glow 590. Ro for ATTO488 and MemGlow 590 was determined as 5.68 nm, calculated from optical and spectral information of the two dyes (*Wu and Brand, 1994*). Then, we went through the above procedure and

estimated the effective Ro for ATTO 488 and PlasMem Bright Red to be 6.45 nm. Note that this value is only valid in our experimental setting, since the molecular information for PlasMem Bright Red was unclear and its surface density on cells was not measurable. Due to this uncertainty, upon the estimation and the use of effective Ro value, we needed to set 'effective surface density' for PlasmMem Bright Red for FLIM-FRET measurement, which is equal to the actual density of MemGlow 590 in cells stained with designated dye concentrations. Under this assumption, the effective Ro was estimated from the regressions in the plot in *Figure 4—figure supplement 1A*, and this effective value was applied to all other FLIM-FRET analyses. We confirmed by flow cytometry that fluorescence signals of bound PlasMem Bright Red to cells linearly increased as MemGlow 590, when both days were added to cell suspension in staining procedure at the equal molar ratios. Therefore, the actual surface density for PlasMem Bright Red should be linearly varied from these effective values.

FLIM-FRET data for different molecules were analyzed in a single Bayesian statistical model:

$$Y\_i = \frac{\sigma}{(\sigma + \alpha\_i)}$$

$$\alpha\_i = \frac{2}{\pi} \frac{1}{Ro^6} z\_i^4,$$

by setting the acceptor density σ as variable, and inferring α_i for each molecule to calculate z_i, which is the Forster distance for each molecule, while $(2/\pi)/(1/Ro^6)$ is constant through all molecules. α_i were bound in the half normal distribution whose variance μ were bound in Gaussian prior distribution, and the Gaussian noise *sc* was added to Y (*Figure 4—figure supplement 1C*). Markov chain Monte Carlo sampling was repeated four times in the scheme of probabilistic programming language NumPyro. Once convergence was confirmed, the protein-specific z_i was determined from the posterior distributions of α_i (*Figure 4—figure supplement 1D*).

## Protein expression and biochemical analysis

cDNA for N-terminus dual-Strep-TagII / SNAP tagged and C-terminus His10 tagged ectodomains of human MUC1 (14TR) and CD43 genes were cloned into pAcGFP-n1 vector (Takara), and cDNA for N-terminus SNAP tagged and C-terminus His10 tagged ectodomains of human MUC1 (14TR) and CD43 genes were cloned into pGEX-6p1 vector (Cytiva). Mammalian expression vectors were transfected into HEK293T cells, and after 48 hr of transfections, cell culture medium supernatant was collected. After filtering, the supernatant containing secreted glycoproteins was concentrated more than 10 times by using Amicon Ultra (Merck Millipore), and then subjected to Strep-tactin column (iba). Trapped glycoproteins were collected by elution from the column. Bacterial expression vectors were introduced into *E. coli* BL21, IPTG-induced at 4 °C for overnight, and then affinity purified using GST tag and GST-bind resin (Novagen) followed by cleavage with PreScission protease (GE Healthcare). These proteins were then stained with 5 µM SNAP – Surface 488 or SNAP –Cell TMR Star and cleaned by Micro Biospin 6. C-terminus His-tagged human EPHB1 ectodomain was obtained from Sino Biological and labeled with ATTO 488 NHS-ester. Protein concentrations were measured by Bradford assay, and the concentration of fluorescent dyes in protein solution was measured by using standard ATTO 488 dyes with known concentrations, by a standard plate reader. In SDS-PAGE analysis, fluorescent and bright field images of gels were obtained using LumiCube Plus imager (Liponics, Tokyo, Japan) and glycan staining was performed by using Pierce Glycoprotein Staining Kit (Thermo Fisher).

## Analysis using Alpha Fold 2

Alpha Fold 2 structural prediction data for all 2515 proteins used for glycosylation analyses were obtained from UniProt. From coordinates of atoms except for hydrogen, the centroid of amino acid at two ends of ectodomain was calculated, and the distance between these two amino acids was calculated as the length of ectodomain in the predicted conformation for all the proteins. In parallel, the longest distance among all amino acids in the ectodomain from the first amino acid in the ectodomain next to the transmembrane domain was calculated. In addition, predicted local Distance Difference test (plDDT) (*Mariani et al., 2013*), which was used as the score to assess Alpha Fold 2 predictions, was also obtained from the database. MUC1 (0TR) was not a natural protein and its predicted conformation was not in the database. Therefore, we input its sequence into Alpha Fold 2 (ver 2.2 in Google Colaboratory) and obtained the predicted conformation.

## In vitro reconstitution on lipid bilayers

Silica beads (4.63 µm diameter, Bangs Laboratory) were coated with lipid bilayers, in a similar manner as described previously (*Lee et al., 2021*). Beads were treated for 30 min by tenfold volume of Piranha solution (3:1 mix of sulfuric acid and hydrogen peroxide) and rinsed extensively by water, and then ~0.2 g/cm³ of cleaned beads were incubated in PBS with small unilamellar vesicles (SUVs), by adjusting lipid concentration ~1 mM in this mixture. SUV contained 10 mol% of 1,2-dioleoyl-sn-glycero-3- [(N-(5-amino-1-carboxypentyl)iminodiacetic acid)succinyl] (DGS-NTA(Ni), Avanti Polar lipids), 20 mol% of 1,2-dioleoyl-sn-glycero-3-phosphatidylserine (DOPS, Avanti Polar lipids), and 70 mol% of 1,2-Diole oyl-sn-glycero-3-phosphatidylcholine (DOPC, Avanti Polar lipids). For nonfluid bilayers, DOPC was replaced with DPPC (1,2-dipalmitoyl-sn-glycero-3-phosphocholine, Avanti Polar lipids). If necessary, 0.02% of tetramethylrhodamine thiocarbamoyl-1,2-dihexadecanoyl-sn-glycero-3-phosphoeth- anol-amine, triethylammonium salt (TRITC-DHPE, Thermo) was also included. SUVs were prepared by soni-cation of 1 mL lipid suspension in PBS (final 1.8 mM total lipids). After the washout of excess SUVs, bilayer-coated beads were incubated with proteins of designated concentrations and subjected to flow cytometry measurement for protein binding to single beads. Surface densities for proteins on beads were measured by flow cytometry and calibrated as described above for measuring densities in cells. Plots for mean surface densities of proteins *d* vs protein concentration *x* in solution were regressed to the standard receptor saturation binding model, $d=Bx/(K_D +x)$, where $K_D$ is the dissociation constant and *B* is the saturated density.

For single molecule tracking experiments, planar supported lipid bilayers were created on cover glass pre-etched by Piranha solution. SNAP –Cell TMR star (New England Biolabs) labeled proteins were incubated with bilayers at 10 pM concentrations, and the dynamics of sparsely distributed proteins on bilayers were imaged by total internal reflection microscopy imaging at 10 fps (Leica AF6000LX equipped with a Cascade II EMCCD camera (Roper)). Images were analyzed by Track Mate plugin of Fiji/ImageJ.

## STORM

Nikon N-STORM system with ECLIPSE Ti2-E equipped with Andor iXon3 EMCCD camera was used for STORM imaging. Cells expressing various SNAP-tagged membrane proteins were fixed and stained with SNAP – surface Alexa 647 and anti-SARS-CoV/SARS-CoV-2 (COVID-19) Spike monoclonal anti-body (clone 1A9) and CF568 labeled anti-Mouse IgG secondary antibody. Pairs of mCherry protein tagged for Lck10 and Alexa 647-labeled anti-Mouse IgG secondary antibody were also used. Stained cells were loaded on glass bottom chamber (Lab-Tek II, Nunc) precoated with 1% PEI, and then cells were immersed in STORM imaging buffer, 50 mM Tris (pH 8.0)–10 mM NaCl buffer containing 10% Glycerol, 0.56 mg/mL Glucose oxidase, 34 µg/mL Catalase, 0.1 M Cysteamine. Both Alexa 647 and CF568 dyes were reactivated continuously by 405 nm laser and imaged by using 561 nm and 647 nm excitation lasers.

Coordinates for centroids of all detected fluorescent STORM spots were collected. The window for image analysis was determined as with 10% margin for longest distance between detected spots to both x- and y- directions. Viral protein spots not close to cell membranes were eliminated by thresh-olding with nearby spot density for cell protein. Specifically, the entire image was pixelated with a 0.5 µm square box, and all viral protein signals within the box that had no membrane protein signals were removed. Also, viral protein spots only sparsely located were eliminated by thresholding with nearby spot density for viral protein. This thresholding process removed any detected viral protein spot that did not have more than 100 other viral protein spots within 1 µm.

For coordinates of spots remained after these thresholding processes, cross-correlation function in a defined size of image window can be described in a discrete form as follows, which is equivalent to the average localization density in the whole field of view (*Schnitzbauer et al., 2018*):

$$C\left(r, \Delta r\right) = \frac{XY}{A\left(r, \Delta r\right) N_a N_b} * H_{AB}\left(r, \Delta r\right),$$

where X and Y are the size of image window in both axes, Na, Nb are the number of spots for each kind of protein in the whole window, $A(r, \Delta r) = \pi \Delta r (2 r + \Delta r)$ is the area of a shell with inner radius *r* and outer radius $r + \Delta r$, and $H_{AB}(r, \Delta r)$ is a histogram of mutual distance between two kinds of dyes (A, B) in a whole image. $\Delta r$, a size of shell and the bin size for histogram, was set to be 5 nm in our analyses.

In the simulated images, the cell membrane is generated as a shell 10 µm in diameter and 100 nm wide, consisting of 10,000 spots of randomly generated membrane proteins within the shell. One virus image consists of 200 random spots within a circle 100 nm in diameter, with one image containing one virus. The coordinates of the center of the virus were varied among the simulated images. $C(r)$ for these simulated images was computed as above; a plot of $C(r)$ versus density_(out/in) showed that a polynomial was a reasonable fit, so Bayesian inference was performed with a fifth-order polynomial.

## Materials availability

All unique/stable reagents generated in this study are available from the corresponding author with a completed materials transfer agreement.

## Acknowledgements

STORM experiment was supported by Imaging Core Laboratory, The Instituteof Medical Science, The University of Tokyo. We thank Kazuaki Tokunaga (Nikon) for his help in STORM sample preparation and imaging, and Hidenobu Nakao (NIMS) for helpful discussion. This work has been supported by intramural funding from NIMS.

## Additional information

### Funding

No external funding was received for this work.

### Author contributions

Yoshihisa Kaizuka, Conceptualization, Resources, Data curation, Software, Formal analysis, Supervision, Funding acquisition, Validation, Investigation, Visualization, Methodology, Writing – original draft, Writing – review and editing; Rika Machida, Data curation, Investigation, Methodology, Writing – review and editing

### Author ORCIDs

Yoshihisa Kaizuka https://orcid.org/0000-0002-8019-0873

Joint Public Review: https://doi.org/10.7554/eLife.101175.3.sa1
Author response https://doi.org/10.7554/eLife.101175.3.sa2

## Additional files

### Supplementary files

Supplementary file 1. List of all reagents and software used in this study.

Supplementary file 2. List of 2515List human membrane proteins whose ectodomain amino sequences were used for glycosylation predictions by NetN/O Glyc and GlycoEP. Length of amino acid sequences in ectodomains as well as ratio of glycosylation site per amino acid are listed for all molecules.

Supplementary file 3. Glycosylation prediction data for MUC1 truncation mutants and mouse proteins used in this study.

MDAR checklist

### Data availability

Original code as well as raw and processed data analyzed through the code are available in an online repository (https://github.com/ykaizuka/eLife_glycoprotein-virus copy archived at *Kaizuka, 2024*). Additional information required to reanalyze the data reported in this paper is available at https://doi.org/10.6084/m9.figshare.30314866.

The following dataset was generated:

| Author(s) | Year | Dataset title | Dataset URL | Database and Identifier |
|---|---|---|---|---|
| Kaizuka Y, Machida R | 2025 | KAIZUKA et al 2025 eLife | https://doi.org/10.6084/m9.figshare.30314866 | figshare, 10.6084/m9.figshare.30314866 |

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
