## [Editor Report · eLife Assessment]

This **fundamental** work substantially advances our understanding of how the glycocalyx of cells provide a non-specific barrier for the interaction of viruses with cell-surface receptors. Using both in vitro experiments and in vivo manipulations they provide **compelling** evidence for the properties of the glycocalyx to serve as an energy barrier as a main attribute of its mode of action. The work will be of broad interest to virologists and the cell biology community that studies host-pathogen interactions.

---

## [Referee Report · Joint Public Review]

This manuscript tests the notion that bulky membrane glycoproteins suppress viral infection through non-specific interactions. Using a suite of biochemical, biophysical, and computational methods in multiple contexts (ex vivo, in vitro, and in silico), the authors collect compelling evidence supporting the notion that (1) a wide range of surface glycoproteins erect an energy barrier for the virus to form stable adhesive interface needed for fusion and uptake and (2) the total amount of glycan, independent of their molecular identity, additively enhanced the suppression.

As a functional assay the authors focus on viral infection starting from the assumption that a physical boundary modulated by overexpressing a protein-of-interest could prevent viral entry and subsequent infection. Here they find that glycan content (measured using the PNA lectin) of the overexpressed protein and total molecular weight, that includes amino acid weight and the glycan weight, is negatively correlated with viral infection. They continue to demonstrate that it is in effect the total glycan content, using a variety of lectin labelling, that is responsible for reduced infection in cells. Because the authors do not find a loss in virus binding this allows them to hypothesize that the glycan content presents a barrier for the stable membrane-membrane contact between virus and cell. They subsequently set out to determine the effective radius of the proteins at the membrane and demonstrate that on a supported lipid bilayer the glycosylated proteins do not transition from the mushroom to the brush regime at the densities used. Finally, using Super Resolution microscopy they find that above an effective radius of 5 nm proteins are excluded from the virus-cell interface.

The experimental design does not present major concerns and the results provide insight on a biophysical mechanism according to which, repulsion forces between branched glycan chains of highly glycosylated proteins exert a kinetic energy barrier that limits the formation of a membrane/viral interface required for infection.

In their revised manuscript and rebuttal, the authors address several general and specific concerns that were raised about their first submission. The revised manuscript now makes the strength of the evidence supporting their claims, compelling.

---

## [Author Response]

The following is the authors’ response to the original reviews.

**Public Review**
GENERAL QUESTIONS:(1) For many enveloped viruses, the attachment factors - paradoxically - are also surface glycoproteins, often complexed with a distinct fusion protein. The authors note here that the glycoportiens do not inhibit the initial binding, but only limit the stability of the adhesive interface needed for subsequent membrane fusion and viral uptake. How these antagonistic tendencies might play out should be discussed.

When the surface density of receptor molecules for a virus with glycans increases, the density of free glycans not bound to the virus increases along with the amount of virus adsorbed. However, if the total amount of glycans is considered to be a function of the receptor density, the reaction may become more complicated. This complication may also be affected by the prolonged infection. If the receptor density on the cell surface is high, the infection inhibitory effect of glycans may not be obtained in a system in which a high concentration of virus is supplied from the outside world for a long time. This is because once viruses have entered the cell, they accumulate inside the cell, and viral infection is affected by the total accumulated amount, which is the integration of the number of viruses that have entered over time. This distinction indicates that the virus entry reaction and the total amount of infection in the cell must be considered separately. This is an important point, but it was not clearly mentioned in the original manuscript.

Our experiments were conducted under conditions that clearly allowed us to detect the virusinhibiting function of glycans without being affected by the above points. In order to clarify these points, we will revise this article as follows, referring to an experiment that is somewhat related to this discussion (the Adenovirus infection experiment into HEK293T cells shown in Figure S1F)..

(Page-3, Introduction)

While there are known examples of glycans that function as viral receptors (Thompson et al., 2019), these results demonstrate that a variety of glycoproteins negatively regulate viral infection in a wide range of systems. All of these results suggest that bulky membrane glycoproteins nonspecifically inhibit viral infection.

(Page 20, Discussion)

When the virus receptor is a glycoprotein or glycan itself, the inhibition of virus infection by glycans becomes more complex because the total amount of glycans is also a function of the receptor density. It is also important to note that the total amount of infection into a cell is the time integral of virus entry. Even if the probability of virus entry is significantly reduced by glycans, the cumulative number of virus entries may increase if high concentrations of virus continue to be supplied from outside the cell for a long period of time. In the case of Adenovirus, which continues to amplify in HEK293T cells after infection, we showed that MUC1 on the cell surface has an inhibitory effect on long-term cumulative infection (Supplementary Figure 1F). However, such an accumulation effect may be caseby-case depending on the virus cell system, and may be more pronounced when the cell surface density of virus receptor molecules is high. As a result, if the virus receptor molecule is a glycan or glycoprotein and infection continues for a long period of time, the infection inhibition effect may not be observed despite an apparent increase in the total amount of glycans in the cell. In any case, our results clarified the factor of virus entry inhibition dependent on the total amount of glycans because appropriate conditions were set.

(2) Unlike polymers tethered to solid surface undergoing mushroom-to-brush transition in densitydependent manner, the glycoproteins at the cell surface are of course mobile (presumably in a density-dependent manner). They can thus redistribute in spatial patterns, which serve to minimize the free energy. I suggest the authors explicitly address how these considerations influence the in vitro reconstitution assays seeking to assess the glycosylation-dependent protein packing.

We performed additional experiments using lipid bilayers that had lost fluidity, and found that there is no significant difference in protein binding between fluid and nonfluid bilayers. The redistribution of molecules due to molecular fluidity may play some roles but not in our experimental systems. It suggests that glycoproteins can generate intermolecular repulsion even in fluid conditions such as cell membranes, just as they do on the solid phase. This experiment was also very useful because it allowed us to compare our results in the fluid bilayer with solid-state measurements of saturation molecular density and the brush transition. This comparison gave us confidence that in the reconstituted membrane system, even at saturation density, the membrane proteins are not as stretched as they are in the condensed brush state. We have therefore added a new paragraph and a new figure (Supplementary Fig. 5B) to discuss this issue, as follows:

The molecular structural state of these proteins needs to be further discussed to estimate the contribution of f_el_, which represents resistance to molecular elongation. Our results suggest that these densely packed nonglycosylated molecules are no longer in a free mushroom state. However, their saturation density was several times lower than previously reported brush transition densities, such as 65000 µm^-2^ for 17 kDa polyacrylamide (R_F_ ~ 15 nm) on a solid surface (Wu et al., 2002). To compare our data on fluid bilayers with previously reported data on solid surfaces, we performed additional experiments with lipid bilayers that lost fluidity. No significant changes in protein binding between fluid and nonfluid bilayers were observed for both b-MUC1 and g-MUC1 molecules (Supplementary Figure 5B). This result suggests that membrane fluidity does not affect the average intermolecular distance or other relevant parameters that control molecular binding in the reconstituted system. Based on these, we speculate that the saturated protein density observed in our experiments is lower than or at most comparable to the actual brush transition density. Thus, although these crowded proteins may be restricted from free random motion, they are not significantly extended as in the condensed brush state, in which the contribution of resistance to molecular extension f_el_ is expected to be small relative to the overall free energy of the system.

(3) The discussion of the role of excluded volume in steric repulsion between glycoprotein needs clarification. As presented, it's unclear what the role of "excluded volume" effects is in driving steric repulsion? Do the authors imply depletion forces? Or the volume unavailable due to stochastic configurations of gaussian chains? How does the formalism apply to branched membrane glycoproteins is not immediately obvious.

Regarding the excluded volume due to steric repulsion between glycoproteins, we considered the volume that cannot be used by glycans as Gaussian chains branching from the main chain. We would like to expand on this point by adding several papers that make similar arguments. I'm glad you brought this up because we hadn't considered depletion forces - the excluded volume between glycoproteins should generate a depletion force, but in this case we believe this force will not have a significant effect on viruses that are larger than the glycoproteins. We also attempted to clarify the discussion in this section by focusing on intermolecular repulsion, and restructured paragraphs, which are also related to General Question 2 and Specific Question 2. The relevant part has been revised as follows. (page 15~page16)：

To compare the packing of proteins with different molecular weights and R_F_, These were smaller than the coverage of molecules at hexagonal close packing that is ~90.7%. In contrast, the coverage of b-CD43 and b-MUC1 at saturated binding was estimated to be greater than 100% under this normalization standard, indicating that the mean projected sizes of these molecules in surface direction were smaller than those expected from their R_F_ Thus, it is clear that glycosylation reduces the saturation density of membrane proteins, regardless of molecular size.

Highly glycosylated proteins resisted densification, indicating that some intermolecular repulsion is occurring. In the framework of polymer brush theory, the intermolecular repulsion of densely packed highly glycosylated proteins is due to an increase in either f_el_, f_int_ (d<R_F_), or both (Hansen et al., 2003; Wu et al., 2002). The term of intermolecular interaction, f_int_, is regulated by intermolecular steric repulsion, which occurs when neighboring molecules cannot approach the excluded volume created by the stochastic configuration of the polymer chain (Attili et al., 2012; Faivre et al., 2018; Kreussling and Ullman, 1954; Kuo et al., 2018; Paturej et al., 2016). The magnitude of this steric repulsion depends largely on R_F_ in dilute solutions, but the molecular structure may also affect it when molecules are densified on a surface. In other words, the glycans protruding between molecules can cause steric inhibition between neighboring proteins (Figure 5D). Such intermolecular repulsion due to branched side chains occurs only when the molecules are in close proximity and sterically interact on a twodimensional surface, but not in dilute solution, and does not occur in unbranched polymers such as underglycosylated proteins (Figure 5D). Based on the above, we propose the following model for membrane proteins: Only when the membrane proteins are glycosylated does strong steric repulsion occur between neighboring molecules during the densification process, suppressing densification.

The molecular structural state of these proteins needs to be further discussed to estimate the contribution of f_el_, which represents resistance to molecular elongation. Our results suggest that these densely packed nonglycosylated molecules are no longer in a free mushroom state. However, their saturation density was several times lower than previously reported brush transition densities, such as 65000 µm^-2^ for 17 kDa polyacrylamide (R_F_ ~ 15 nm) on a solid surface (Wu et al., 2002). To compare our data on fluid bilayers with previously reported data on solid surfaces, we performed additional experiments with lipid bilayers that lost fluidity. No significant changes in protein binding between fluid and nonfluid bilayers were observed for both b-MUC1 and g-MUC1 molecules (Supplementary Figure 5B). This result suggests that membrane fluidity does not affect the average intermolecular distance or other relevant parameters that control molecular binding in the reconstituted system. Based on these, we speculate that the saturated protein density observed in our experiments is lower than or at most comparable to the actual brush transition density. Thus, although these crowded proteins may be restricted from free random motion, they are not significantly extended as in the condensed brush state, in which the contribution of resistance to molecular extension f_el_, is expected to be small relative to the overall free energy of the system.

Note that this does not mean that glycoproteins cannot form condensed brush structures: in fact, highly glycosylated molecules (e.g., MUC1) can form brush structures in cells when such proteins are expressed at very high densities. (Shurer et al., 2019). In these cells, ………. Such membrane deformation results in the increase of total surface area to reduce the density of glycoproteins, indicating that there is strong intermolecular repulsion between glycoproteins. In any case, the free energy of the system is determined by the balance between protein binding and insertion into the membrane, protein deformation, and repulsive forces between proteins, which determine the density of proteins depending on the configuration of the system. Thus, although strong intermolecular repulsions were prominently observed in our simplified system, this may not be the case in other systems. ……

(4) The authors showed that glycoprotein expression inversely correlated with viral infection and link viral entry inhibition to steric hindrance caused by the glycoprotein. Alternative explanations would be that the glycoprotein expression (a) reroutes endocytosed viral particles or (b) lowers cellular endocytic rates and via either mechanism reduce viral infection. The authors should provide evidence that these alternatives are not occurring in their system. They could for example experimentally test whether non-specific endocytosis is still operational at similar levels, measured with fluid-phase markers such as 10kDa dextrans.

The results of the experiment suggested by the reviewer are shown in the new Supplementary Figure 3B. (This results in generation of a new Supplementary Figure 3, and previous Supplementary Figures 4-5 are now renumbered as Supplementary Figures 5-6). Endocytosis of 10KDa dextran was attenuated by the expression of several large-sized molecules, but was not affected by the expression of many other glycoproteins that have the ability to inhibit infection. These results were clearly different from the results in which virus infection was inhibited more by the amount of glycan than by molecular weight. Therefore, it was found that many glycoproteins inhibit virus infection through processes other than endocytosis. Based on the above, we have added the following to the manuscript: （p9 New paragraph:）

We also investigated the effect of membrane glycoproteins on membrane trafficking, another process involved in viral infection. Expression of MUC1 with higher number of tandem repeats reduced the dextran transport in the fluid phase, while expression of multiple membrane glycoproteins that have infection inhibitory effects, including truncated MUC1 molecules, showed no effect on fluid phase endocytosis, indicating a molecular weight-dependent effect (Supplementary Figure 3B). The molecular weight-dependent inhibition of endocytosis may be due to factors such as steric inhibition of the approach of dextran molecules or a reduction in the transportable volume within the endosome. In any case, it is clear that many low molecular weight glycoproteins inhibit infection by disturbing processes other than endocytosis. Based on the above, we focus on the effect of glycoproteins on the formation of the interface between the virus and the cell membrane.

(5) The authors approach their system with the goal of generalizing the cell membrane (the cumulative effect of all cell membrane molecules on viral entry), but what about the inverse? How does the nature of the molecule seeking entry affect the interface? For example, a lipid nanoparticle vs a virus with a short virus-cell distance vs a virus with a large virus-cell distance?

Thank you for your interesting comment. If the molecular size of the ligand is large, it should affect virus adsorption and molecular exclusion from the interface. In lipid nanoparticle applications, controlling this parameter may contribute to efficiency. In addition, a related discussion is the influence of virus shell molecules that are not bound to the receptor. I will revise the text based on the above.

Discussion (as a new paragraph after the paragraph added in Q1):

In this study, we attempted to generalize the surface structure on the cell side, but the surface structure on the virus side may also have an effect. The efficiency of virus adsorption and the efficiency of cell membrane protein exclusion from the interface will change depending on the molecular length of the receptor-ligand, although receptor priming also has an effect. In addition, free ligands of the viral envelope or other coexisting glycoproteins may also have an effect as they are also required for exclusion from the virus-cell interface. In fact, there are reports that expression of CD43 and PSGL-1 on the virus surface reduces virus infection efficiency (Murakami et al., 2020). Such interface structure may be one of the factors that determine the infection efficiency that differs depending on the virus strain. More generally, modification of the surface structure may be effective for designing materials such as lipid nanoparticles that construct the interface with cell.

SPECIFIC QUESTIONS:(1) The proposed mechanism indicates that glycosylation status does not produce an effect in the "trapping" of virus, but in later stages of the formation of the virus/membrane interface due to the high energetic costs of displacing highly glycosylated molecules at the vicinity of the virus/membrane interface. It is suggested to present a correlation between the levels of glycans in the Calu-3 cell monolayers and the number of viral particles bound to cell surface at different pulse times. Results may be quantified following the same method as shown in Figure 2 for the correlation between glycosylation levels and viral infection (in this case the resulting output could be number of viral particles bound as a function of glycan content).

The results of this experiment are now shown as Supplementary Figure 2F and 2G. We compared the amount of virus bound after incubation for 10 minutes or for 3 hours as in the infection experiment, but no negative correlation was found between the total amount of glycans on the surface of the Calu3 monolayer and the amount of virus bound. Interestingly, there was a sight positive correlation was detected, which may be due to concentrated virus receptor expressions in glycan-enriched cells. This result shows that glycoproteins do not strongly inhibit virus binding. We will amend the text as follows (see also Q6).

(Page 10)

Glycans could be one of the biochemical substances ……We found that a large number of SARS-CoV2-PP can still bind to cells even when cells expressed sufficient amounts of the glycoprotein that could account for the majority of glycans within these cells and inhibit viral infection (Figure 3A). Similarly, on the two-dimensional culture surface of Calu-3 cells, no negative correlation was observed between the number of viruses bound and the total amount of glycans on the cell surface (Supplementary Figure 2F-G). The slight positive correlation between bound virus and glycans may be due to higher expression levels of viral receptors in glycan-rich cells. ….

(2) The use of the purified glycosylated and non-glycosylated ectodomains of MUC1 and CD-43 to establish a relationship between glycosylation and protein density into lipid bilayers on silica beads is an elegant approach. An assessment of the impact of glycosylation in the structural conformation of both proteins, for instance determining the Flory radius of the glycosylated and non-glycosylated ectodomains by the FRET-FLIM approach used in Figure 4 would serve to further support the hypothesis of the article.

Unfortunately, the proposed experiment did not provide a strong enough FRET signal for analysis. This was due in part to the difficulty in constructing a bead-coated bilayer incorporating PlasMem Bright Red, which established a good FRET pair in cell experiments. We also tried other fluorescent molecules, but were unable to obtain a strong and stable FRET signal. Another reason may be that the curvature of the beads is larger than that of the cells, making it difficult to obtain a sufficient cumulative FRET effect from multiple membrane dyes. We plan to improve the experimental system in the future.

On the other hand, we also found that in this system, the signal changes were very subtle, making it difficult to detect molecular conformational changes using FRET. After reconsidering general questions (2) and (3), we speculated that the molecular density in the experiment, even at saturation binding, was below or at most equivalent to the brush transition point. In other words, proteins on the bead-coated bilayer may not be significantly extended in the vertical direction. Therefore, the conformational changes of these proteins may not be large enough to be detected by the FRET assay. We updated Figure 3C and Figure 5D (model description) to better reflect the above discussion and introduced the following discussion in the manuscript.

(page11)

We introduced the framework of conventional polymer brush theory to study the structure of viruscell interfaces containing proteins……. Numerous experimental measurements of the formation of polymer brushes have also been reported (Overney et al., 1996; Wu et al., 2002; Zhao and Brittain, 2000). In these measurements, the transition to a brush typically occurs at a density higher than that required to pack a surface with hemispherical polymers of diameter R_F_. This is the point at which the energy loss due to repulsive forces between adjacent molecules (f_int_) exceeds the energy required to stretch the polymer perpendicularly into a brush (f_el_).

(page16)

The molecular structural state of these proteins needs to be further discussed to estimate the contribution of f_el_, which represents resistance to molecular elongation. Our results suggest that these densely packed nonglycosylated molecules are no longer in a free mushroom state. However, their saturation density was several times lower than previously reported brush transition densities, such as 65000 µm^-2^ for 17 kDa polyacrylamide (R_F_ ~ 15 nm) on a solid surface (Wu et al., 2002). To compare our data on fluid bilayers with previously reported data on solid surfaces, we performed additional experiments with lipid bilayers that lost fluidity. No significant changes in protein binding between fluid and nonfluid bilayers were observed for both b-MUC1 and g-MUC1 molecules (Supplementary Figure 5B). This result suggests that membrane fluidity does not affect the average intermolecular distance or other relevant parameters that control molecular binding in the reconstituted system. Based on these, we speculate that the saturated protein density observed in our experiments is lower than or at most comparable to the actual brush transition density. Thus, although these crowded proteins may be restricted from free random motion, they are not significantly extended as in the condensed brush state, in which the contribution of resistance to molecular extension f_el_ is expected to be small relative to the overall free energy of the system.

Note that this does not mean that glycoproteins cannot form condensed brush structures: in fact, highly glycosylated molecules (e.g., MUC1) can form brush structures in cells when such proteins are expressed at very high densities. (Shurer et al., 2019). In these cells, ………. Such membrane deformation results in the increase of total surface area to reduce the density of glycoproteins, indicating that there is strong intermolecular repulsion between glycoproteins. In any case, the free energy of the system is determined by the balance between protein binding and insertion into the membrane, protein deformation, and repulsive forces between proteins, which determine the density of proteins depending on the configuration of the system. Thus, although strong intermolecular repulsions were prominently observed in our simplified system, this may not be the case in other systems. ……

(3) The MUC1 glycoprotein is reported to have a dramatic effect in reducing viral infection shown in Fig 1F. On the contrary, in a different experiment shown in Fig2D and Fig2H MUC1 has almost no effect in reducing viral infection. It is not clear how these two findings can be compatible.

The immunostaining results show that the density of MUC1 molecules is very low in the experimental system in Figure 2 (Figure 2C), which is supported by the SC-RNASeq data (as shown in Supplementary Figure 2A, MUC1 is not listed as a top molecule). In other words, the MUC1 expression level in this experiment is too low to affect virus infection inhibition. On the other hand, the Pearson correlation function represents the strength of the linear relationship between two variables, so it is not the most appropriate indicator for seeing the correlation with the MUC1 expression level, which has little change (Figure 2D, 2F). In fact, even TOS analysis, which can see the correlation by focusing on the cells with the highest expression level, cannot detect the correlation (Figure 2H).Therefore, the MUC1 data in Figure 2DFH will be annotated and corrected in the figure legend.

Figure2 Legend:

MUC1 has a small mean expression level and variance, and is more affected by measurement noise than other molecules when calculating the Pearson correlation function (Figure 2C-2F). In addition, the number of cells in which expression can be detected is small, so no significant correlation was detected by TOS analysis (Figure 2H).

(4) Why is there a shift in the use of the glycan marker? How does this affect the conclusions? For the infection correlation relating protein expression with glycan content the PNA-lectin was used together with flow cytometry. For imaging the infection and correlating with glycan content the SSA-lectin is used.

For each cell line, we selected the lectin that could be measured over the widest dynamic range. This lectin is thought to recognize the predominant glycan species in the cell line (Fig. S1C, Fig. 2D). In our model, we believe that viral infection inhibition is not specific to the type of sugar, but is highly dependent on the total amount of glycans. If this hypothesis is correct, the reason we used different lectins in each experiment is simply to select the lectin that recognizes the most predominant glycan species that is most convenient for predicting the total amount of glycans in cells. This hypothesis is consistent with our observations, where the total amount of glycans estimated by different lectins could explain the infection inhibition in a similar way in the experiments in Figures 1 and 2, and the TOS analysis in Figure 2 showed that minor glycans also have an infection inhibitory effect. On the other hand, it is of course possible to predict the total amount of glycans more accurately by obtaining as much information on glycans as possible (related to Q5). Based on the above discussion, the manuscript will be revised as follows.

Page5

Using HEK293T cell lines exogenously expressing genes of these proteins tagged with fluorescent markers, their glycosylation was measured by binding of a lectin from Arachis hypogaea (PNA), and the number of these proteins in the cells was measured simultaneously. PNA was used for the measurement because it has a wider dynamic range than other lectins (Supplementary Figure 1C). This suggests that GalNAc recognized by PNA is predominantly present on glycans of HEK293T cells, especially on the termini of glycans that are amenable to lectin binding, compared to other saccharides.. …

page9

Our findings suggest that membrane glycoproteins nonspecifically inhibit viral infection, and we hypothesize that their inhibitory function is also nonspecific depending on the type of glycan. Our hypothesis is consistent with the observations in the TOS analysis. Although minor saccharide species in the system (such as GlcNAc and GalNAc recognized by DSA, WGA, or PNA) showed anticolocalization with infection, their scores were much lower than those of major saccharide species. This suggests that all major and minor saccharide species have an infection inhibitory effect, but cells enriched with minor type glycans are only partially present in the system, and the contribution of these cells to virus inhibition is also partial. It is also consistent with the observation that the amount of GalNAc recognized by PNA determines the virus infection inhibition in HEK 293T cells (Figure 1). Therefore, we believe that our assay using a single type of predominantly expressed lectin is still useful for estimating the total glycan content. Nevertheless, the virus infection rate may show a better correlation with a more accurately estimated total glycan in each cell. For example, the use of multiple lectins with appropriate calibration to integrate multiple signals to simultaneously detect a wider range of saccharide species would allow for more accurate estimation. It should be noted that the amount of bound lectin does not necessarily measure the overall glycan composition but likely reflects the sugar population at the free end of the glycan chain to which the lectin binds most.

(5) The authors in several instances comment on the relevance and importance of the total glycan content. Nevertheless, these conclusions are often drawn when using only one glycan-binding lectin. In fact, the anti-correlation with viral infection is distinct for the various lectins (Fig 2D and Fig 2H). Would it make more sense to use a combination of lectins to get a full glycan spectrum?

As stated in the answer to Q4, we believe that we were able to detect the infection-suppressing effect of the total glycan amount by using the measurement value of the major component glycan as an approximation. However, as you pointed out, if we could accurately measure the minor glycan components and add up their values, we believe that we could measure the total glycan amount more accurately. In order to measure multiple glycans simultaneously and with high accuracy, some kind of biochemical calibration may be necessary to compare the measurements of lectin-glycan pairs with different binding constants. We believe that these are very useful techniques, and would like to consider them as a future challenge. The corrections listed in Q4 are shown below.

(Page 9)

Nevertheless, the virus infection rate may show a better correlation with a more accurately estimated total glycan in each cell. For example, the use of multiple lectins with appropriate calibration to integrate multiple signals to simultaneously detect a wider range of glycans would allow for more accurate estimation. …….

(6) Fig 3A shows virus binding to HEK cells upon MUC1 expression. Please provide the surface expression of the MUC1 so that the data can be compared to Fig 1F. Nevertheless, it is not clear why the authors used MUC expression as a parameter to assess virus binding. Alternatively, more conclusive data supporting the hypothesis would be the absence of a correlation between total glycan content and virus binding capacity.

The relationship between the expression level of MUC1 in each cell and the amount of virus binding is shown in Supplementary Figure 3A. There is no correlation between the two. In HEK293T cells, many glycans are modified with MUC1, so MUC1 was used as the indicator for analysis (Supplementary Figure 1C). As you pointed out, it is better to use the amount of glycan as an indicator, so we analyzed the relationship between the amount of bound virus and the amount of glycan on the surface on the Calu-3 monolayer (Supplementary Figure 2F, 2G, introduced in the answer to Specific (Q1)). In any case, no correlation was found between virus binding and surface glycans. I will correct the manuscript as follows.

(page 9)

Glycans could be one of the biochemical substances that link the intracellular molecular composition and macroscopic steric forces at the cell surface. To clarify this connection, we further investigated the mechanism by which membrane glycoproteins inhibit viral infection. First, we measured viral binding to cells to determine which step of infection is inhibited. We found that a large number of SARS-CoV2-PP can still bind to cells even when cells expressed sufficient amounts of the glycoprotein that could account for the majority of glycans within these cells and inhibit viral infection (Figure 3A). Similarly, on the two-dimensional culture surface of Calu-3 cells, no correlation was observed between the number of viruses bound and the total amount of glycans on the cell surface (Supplementary Figure 2F-G). These results indicate that glycoproteins do not inhibit virus binding to cells, but rather inhibit the steps required for subsequent virus internalization.

(7) While the use of the Flory model could provide a simplification for a (disordered) flexible structure such as MUC1, where the number of amino acids equals N in the Flory model, this generalisation will not hold for all the proteins. Because folding will dramatically change the effective polypeptide chain-length and reduce available positioning of the amino acids, something the authors clearly measured (Fig 4G), this generalisation is not correct. In fact, the generalisation does not seem to be required because the authors provide an estimation for the effective Flory radius using their FRET approach

Current theories generalizing the Flory model to proteins are incomplete, and it is certainly not possible to accurately estimate the size of individual molecules undergoing different folding. However, we found such a generalized model to be useful in understanding the overall properties of membrane proteins. In our experiments, we were indeed able to obtain the R_F_s of some individual molecules by FRET measurements. However, this modeling made it possible to estimate the distribution range of the RFs, including for larger proteins that cannot be measured by FRET. For example, from our results, we can estimate that the upper limit of the RFs of the longest membrane proteins is about 10.5 nm, assuming that the proteins follow the Flory model in all respects except for the shortening of the effective length due to folding. These analyses are useful for physical modeling of nonspecific phenomena, as in our case.

In order to discuss the balance between such theoretical validity and the convenience of practical handling, we revise the manuscript as follows.

(page 13)

This shift in ν indicates that glycosylation increases the size of the protein at equilibrium, but the change in R_F_ is slight, e.g., a 1.3-fold increase for one of the longest ectodomains with N = 4000 when these values of ν are applied. This calculation also gives a rough estimate of the upper limit of the R_F_ of the extracellular domains of all membrane proteins in the human genome (approximately 10.5 nm). Physically, this change in ν by glycosylation may be caused by the increased intramolecular exclusion induced sterically between glycan chains. This estimated ν are much smaller than that of 0.6 for polymers in good solvents, possibly due to protein folding or anchoring effects on the membrane. In fact, the ν of an intrinsically disordered protein in solution has been reported to be close to 0.6 (Riback et al., 2019; Tesei et al., 2024). Overall, these analyses using the Flory model provide information on the size distribution of membrane proteins and the influence of glycans, although the model cannot predict the exact size of each protein due to its specific folding.

MINOR COMMENTS/EDITS:(1) In Figures 2A and 2C, as well as Supplemental Figure 2C, the fluorescent images indicate that GFP expression differs among the various groups. Ideally, these should be at the same GFP expression level, as the glycan and antibody staining occurred post-viral infection. For instance, ACE2 is a well-known positive control and should enhance SARS-CoV-2 infection. Yet, based on the findings presented in Supplemental Figure 2C, ACE2 appears to correlate with the lowest infection rate. The relationship between the infection rate and key glycoproteins needs clearer quantification.

We measured the virus inhibition effect specific to each molecule using a cell line expressing low levels of viral receptors and glycoproteins (Fig. 1). On the other hand, the system in Fig. 2 contains diverse viral receptors and glycoproteins and has not been genetically manipulated. (We apologize that there was a typo in our description of experiment, which will be corrected, as shown below). The variation in infection rate between samples was caused by multiple factors but was not related to the molecule for which the correlation was measured. The receptor-based normalization used in the experiment in Fig. 1 cannot be applied in this system in Fig.2 due to the complexity of the gene expression profile. Therefore, instead of such parameter-based normalization, we applied Pearson correlation and TOS analysis. In the calculation of Pearson correlation, intensities are normalized. TOS analysis allows the analysis of colocalization between the groups with the highest fluorescence intensity. Therefore, in both cases of variation in overall infection rate and variation in the distribution of infected populations, samples with large variations can be reasonably compared by Pearson correlation and TOS analysis, respectively. We extend the discussion on statistics and revise the manuscript as follows.

(page 8-9)

To test this hypothesis, we infected a monolayer of epithelial cells endogenously expressing highly heterogeneous populations of glycoproteins with SARS-CoV-2-PP, and measured viral infection from cell to cell visually by microscope imaging. …

Pearson correlation is effective for comparing samples with varying scales of data because it normalizes the data values by the mean and variance. However, as observed in our experiments, this may not be the case when the distribution of data within a sample varies between samples. In addition, as has already been reported, the distribution of infected cells often deviates significantly from the normal distribution of data that is the premise of Pearson correlation (Russell et al., 2018) (Figure 2B). To further analyze data in such nonlinear situations, we applied the threshold overlap score (TOS) analysis (Figure 2G-H, Supplementary Figure 2E). This is one statistical method for analyzing nonlinear correlations, and is specialized for colocalization analysis in dual color images (Sheng et al., 2016). TOS analysis involves segmentation of the data based on signal intensity, as in other nonlinear statistics (Reshef et al., 2011). The computed TOS matrix indicates whether the number of objects classified in each region is higher or lower than expected for uniformly distributed data, which reflects co-localization or anti-localization in dual-color imaging data. For example, calculated TOS matrices show strong anti-localization for infection and glycosylation when both signals are high (Figure 2GH). This confirms that high infection is very unlikely to occur in cells that express high levels of glycans. The TOS analysis also yielded better anti-localization scores for some of the individual membrane proteins, especially those that are heterogeneously distributed across cells (Figure 2H). This suggests that TOS analysis can highlight the inhibitory function of molecules that are sparsely expressed among cells, reaffirming that high expression of a single type of glycoprotein can create an infection-protective surface in a single cell and that such infection inhibition is not protein-specific. In contrast, for more uniformly distributed proteins such as the viral receptor ACE2, TOS analysis and Pearson correlation showed similar trends, although the two are mathematically different (Figure 2D, 2H). Because glycoprotein expression levels and virus-derived GFP levels were treated symmetrically in these statistical calculations, the same logic can be applied when considering the heterogeneity of infection levels among cells. Therefore, it is expected that TOS analysis can reasonably compare samples with different virus infection level distributions by focusing on cells with high infection levels in all samples.

(2) For clarity, the authors should consider separating introductory and interpretive remarks from the presentation of results. These seem to get mixed up. The introduction section could be expanded to include more details about glycoproteins, their relevance to viral infection, and explanations of N- and O-glycosylation.

Following the suggestion, (1) we added an explanation of the relationship between glycoproteins and viral infection, and N-glycosylation and O-glycosylation to the Introduction section, and (2) moved the introductory parts in the Results section to the Introduction section, as follows.

（１; page3）

While there are known examples of glycans that function as viral receptors (Thompson et al., 2019), these results demonstrate that a variety of glycoproteins negatively regulate viral infection in a wide range of systems. These glycoprotein groups have no common amino acid sequences or domains. The glycans modified by these proteins include both the N-type, which binds to asparagine, and the O-type, which binds to serine and threonine. Furthermore, there have been no reports of infection-suppressing effects according to the specific monosaccharide type in the glycan. All of these results suggest that bulky membrane glycoproteins nonspecifically inhibit viral infection.

(2 : Page 4-5)

To confirm that glycans are a general chemical factor of steric repulsion, an extensive list of glycoproteins on the cell membrane surface would be useful. The wider the range of proteins to be measured, the better. Therefore, we collect information on glycoproteins on the genome and compile them into a list that is easy to use for various purposes. Then, by analyzing sample molecules selected from this list, it may be possible to infer the effect of the entire glycoprotein population on the steric inhibition of virus infection, despite the complexity and diversity of the Glycome (Dworkin et al., 2022; Huang et al., 2021; Moremen et al., 2012; Rademacher et al., 1988). Elucidation of the mechanism of how glycans regulate steric repulsion will also be useful to quantitatively discuss the relationship between steric repulsion and intracellular molecular composition. For this purpose, we apply the theories of polymer physics and interface chemistry.

ResultsList of membrane glycoproteins in human genome and their inhibitory effect on virus infection

To test the hypothesis that glycans contribute to steric repulsion at the cell surface, we first generate a list of glycoproteins in the human genome and then measure the glycan content and inhibitory effect on viral infection of test proteins selected from the list (Figure 1A). To compile the list of glycoproteins, we ….

(3) In the sentence, "glycoproteins expressed lower than CD44 or other membrane proteins including ERBB2 did not exhibit any such correlation, although ERBB2 expressed ~4 folds higher amount than CD44 and shared ~7% among all membrane proteins," it is unclear which protein has a higher expression level: CD44 or ERBB2? Furthermore, the use of the word "although" needs clarification.

Corrected as follows:

(page 8)

……showed a weak inverse correlation with viral infection; even such a weak correlation was not observed with other proteins, including ERBB2, which is approximately four-fold more highly expressed than CD44

(4) In Supplementary Figure 5, please provide an explanation of the data in the figure legend, particularly what the green and red signals represent.

Corrected as follows:

STORM images of all analyzed cells, expressing designated proteins. The detected spots of SNAPsurface Alexa 647 bound to each membrane protein are shown in red, and the spots of CF568conjugated anti-mouse IgG secondary antibody that recognizes Spike on SARS-CoV2-PP are shown in green. For cells, a pair of two-color composite images and a CF658-only image are shown. Numbers on axes are coordinates in nanometer.

(5) It would be good to see a comprehensive demonstration of the exact method for estimation of membrane protein density (in the SI), since this is an integral part of many of the analyses in this paper. The method is detailed in the Methods section in text and is generally acceptable, but this methodology can vary quite widely and would be more convincing with calibration data provided.

We added flow cytometry and fluorometer data for calibration (Supplementary Figure 1L,M) and introduced a sentence explaining the procedure for obtaining the values used for calibration as follows:

(page 54)

…….Liposome standards containing fluorescent molecules (0.01– 0.75 mol% perylene (Sigma), 0.1– 1.25 mol% Bodipy FL (Thermo), and 0.005– 0.1% DiD) as well as DOPC (Avanti polar lipids) were measured in flow cytometry (Supplmentary Figure 1L). Meanwhile, by fluorimeter, fluorescence signals of these liposomes and known concentrations of recombinant mTagBFP2, AcGFP and TagRFP-657 proteins and SNAP-Surface 488 and Alexa 647 dyes (New England Biolabs) were measured in the same excitation and emission ranges as in flow cytometry assays (Supplementary Figure 1M). Ratios between the integral of fluorescent intensities in this range between two dyes of interest are used for converting the signals measured in flow cytometry. Additional information needed for calibration is the size difference between liposomes and cells. The average diameter of liposomes is measured to be 130 nm, and the diameter of HEK 293T cells is estimated to be 13 µm (Furlan et al., 2014; Kaizuka et al., 2021b; Ushiyama et al., 2015). From these data, the signal from cells acquired by flow cytometry can be calibrated to molecular surface density. For example, the Alexa 647 signal acquired by flow cytometry can be converted to the signal of the same concentration of DID dye using fluorometer data, but the density of the dye is unknown at this point. This converted DID signal can then be calibrated to the density on liposomes rather than cells using liposome flow cytometry data. Finally, adjusted for the size difference between liposomes and cells, the surface molecular density on cells is determined. By going through one cycle of these procedures, we could obtain calibration unit, such as 1 flow cytometry signal for a cell in the designated illumination and detection setting = 0.0272 mTagBFP2 µm^-2^ on cell surface.

(Figure legend, Supporting Figure 1:)

… L. Flow cytometry measurements for liposomes containing serially diluted dye-conjugated lipids and fluorescent membrane incorporating molecules (Bodipy-FL, peryelene, and DID) with indicated mol%. Linear fitting shown was used for calibration. M. Fluorescence emission spectrum for equimolar molecules (50µM for green and far-red channels, and 100µM for blue channel), excited at 405 nm, 488 nm, and 638 nm, respectively. Membrane dyes were measured as incorporated in liposomes. Purified recombinant mTagBFP2 was used.

(6) Fig 2A: The figure legend should describe the microscopy method for a quick and easy reference.

Corrected as follows:

(Figure legend, Figure 2)

A. Maximum projection of Z-stack images at 1 µm intervals taken with a confocal microscope. SARSCoV2-pp-infected, air-liquid interface (ALI)-cultured Calu-3 cell monolayers were chemically fixed and imaged by binding of Alexa Fluor 647-labeled Neu5AC-specific lectin from Sambucus sieboldiana (SSA) and GFP expression from the infecting virus.

(7) Fig 2B: what is the color bar supposed to represent? Is it the pixel density per a particular value? Units and additional description are required. In addition, these are "arbitrary units" of fluorescence, but you should tell us if they've been normalized and, if so, how. They must have been normalized, since the values are between 0 and 1, but then why does the scale bar for SSA only go to 0.5?

The color bar shows the number of pixels for each dot, resulting in the scale for density scatter plot. The scale on the X-axis was incorrect. All these issues have been fixed in this revision, in the figure and in the legend as follows.

(Figure legend, Figure 2)

B. Density scatter plot of normalized fluorescence intensities in all pixels in Figure 2A in both GFP and SSA channels. Color indicates the pixel density.

(8) Fig 3D has a typo: this should most likely be "grafted polymer."(9) Fig 3E has a suspected typo: in the text, the author uses the word "exclusion" instead of "extrusion." The former makes more sense in this context.(10) Fig 5A has a typo: "Suppoorted" instead of Supported Lipid Bilayer.(11) Fig 7E-F has a suspected typo: Again, this should most likely be the word "exclusion" instead of "extrusion."

Thank you so much for pointing out these mistakes, I have corrected them all as suggested.

(12) Which other molecules are referred to, on page 6 (middle), that do not have an inhibitory effect? Please specify.

We specified the molecules that have inhibitory effects, and revised as follows:

These proteins include those previously reported (MUC1, CD43) as well as those not yet reported (CD44, SDC1, CD164, F174B, CD24, PODXL) (Delaveris et al., 2020; Murakami et al., 2020). In contrast, other molecules (VCAM-1, EPHB1, TMEM123, etc.) showed little inhibitory effect on infection within the density range we used.

(13) Fig 2 B: the color LUT is not labelled nor explained.

Corrected as described in (7)

(14) Please provide the scale bars for figures Fig 2A, C, E and Suppl Fig 2C, D.

Corrected.

(15) Please provide the name for the example of a 200 aa protein that is meant to inhibit viral infection but is not bigger than ACE2. Also providing the densities in Fig 3A would help to correlate the data to Fig 1F.

Corrected as follows:

(page 10)

We found that a large number of SARS-CoV2-PP can still bind to cells even when cells expressed sufficient amounts of the glycoprotein (mean density ~50 µm^-2^) that could account for the majority of glycans within these cells and inhibit viral infection (Figure 3A). …..

In our measurements, a protein with extracellular domain of ~200 amino acids (e.g. CD164 (138aa)) at a density of ~100 μm-2 showed significant inhibition in viral infection. This molecule is shorter than the receptor ACE2 (722 aa),

(16) In the experiments conducted in HeK cells expressing the different glycoproteins studies it is mentioned that results of infection were normalised by the amount ACE2 expression. Is the expression of receptor homogenous in the experiments conducted in Figure 2? Clarify in the methods if the expression of receptor has been quantified and somehow used to correct the intensity values of GFP used to determine infection.

As also explained for Q1, the system in Fig. 2 contains diverse viral receptors and glycoproteins, and the receptor-based normalization used in the experiment in Fig. 1 cannot be applied. Instead, we applied Pearson correlation and TOS analysis. In the calculation of Pearson correlation, intensities are normalized. TOS analysis allows the analysis of colocalization between the groups with the highest fluorescence intensity. Therefore, in both cases of variation in overall infection rate and variation in the distribution of infected populations, samples with large variations can be reasonably compared by Pearson correlation and TOS analysis, respectively. We extend the discussion on statistics and revise the manuscript as follows.

(page 8-9)

Pearson correlation is effective for comparing samples with varying scales of data because it normalizes the data values by the mean and variance. However, as observed in our experiments, this may not be the case when the distribution of data within a sample varies between samples. In addition, as has already been reported, the distribution of infected cells often deviates significantly from the normal distribution of data that is the premise of Pearson correlation (Russell et al., 2018) (Figure 2B). To further analyze data in such nonlinear situations, we applied the threshold overlap score (TOS) analysis (Figure 2G-H, Supplementary Figure 2E). This is one statistical method for analyzing nonlinear correlations, and is specialized for colocalization analysis in dual color images (Sheng et al., 2016). TOS analysis involves segmentation of the data based on signal intensity, as in other nonlinear statistics (Reshef et al., 2011). The computed TOS matrix indicates whether the number of objects classified in each region is higher or lower than expected for uniformly distributed data, which reflects co-localization or anti-localization in dual-color imaging data. For example, calculated TOS matrices show strong anti-localization for infection and glycosylation when both signals are high (Figure 2GH). This confirms that high infection is very unlikely to occur in cells that express high levels of glycans. The TOS analysis also yielded better anti-localization scores for some of the individual membrane proteins, especially those that are heterogeneously distributed across cells (Figure 2H). This suggests that TOS analysis can highlight the inhibitory function of molecules that are sparsely expressed among cells, reaffirming that high expression of a single type of glycoprotein can create an infection-protective surface in a single cell and that such infection inhibition is not protein-specific. In contrast, for more uniformly distributed proteins such as the viral receptor ACE2, TOS analysis and Pearson correlation showed similar trends, although the two are mathematically different (Figure 2D, 2H). Because glycoprotein expression levels and virus-derived GFP levels were treated symmetrically in these statistical calculations, the same logic can be applied when considering the heterogeneity of infection levels among cells. Therefore, it is expected that TOS analysis can reasonably compare samples with different virus infection level distributions by focusing on cells with high infection levels in all samples.

(17) Can you provide additional details about the method of thresholding to eliminate "background" localisations in STORM?

Method section was corrected as follows:

(page 59)

…Viral protein spots not close to cell membranes were eliminated by thresholding with nearby spot density for cell protein. Specifically, the entire image was pixelated with a 0.5µm square box and all viral protein signals within the box that had no membrane protein signals were removed. Also, viral protein spots only sparsely located were eliminated by thresholding with nearby spot density for viral protein. This thresholding process removed any detected viral protein spot that did not have more than 100 other viral protein spots within 1µm.

(18) The article says "It was shown that the number of bound lectins correlated with the amount of glycans, not with number of proteins (Figure 1E)". Figure 1E correlates experimental PNA/mol with predicted glycosylation sites, not with the number of expressed proteins. Correct sentence with the right Figure reference.

As you pointed out, the meaning of this sentence was not clear. We have amended it as follows to clarify our intention:

(page 8)

Since a wide range of glycoproteins inhibit viral infection, it is possible that all types of glycoproteins have an additive effect for this function. ……. In this cell line, this inverse correlation was most pronounced when quantifying N-acetylneuraminic acid (Neu5AC, recognized by lectins SSA and MAL) compared to the various types of glycans, while some other glycans also showed weak correlations (Supplementary Figure 2C). These results showed that the amount of virus infection in cell anticorrelated with the amount of total glycans on the cell surface. As amount of glycans is determined by the total population of glycocalyx, infection inhibitory effect can be additive by glycoprotein populations as we hypothesized.

If the inhibitory effect is nonspecific and additive, the contribution of each protein is likely to be less significant. To confirm this, we also measured the correlation between the density of each glycoprotein and viral infection. CD44, which was shown to…….. Our results demonstrate that total glycan content is a superior indicator than individual glycoprotein expression for assessing infection inhibition effect generated by cell membrane glycocalyx. These results are consistent with our hypothesis regarding the additive nature of the nonspecific inhibitory effects of each glycoprotein.